# Water coordinated on Cu(I)-based catalysts is the oxygen source in $CO_2$ reduction to CO

Yajun Zheng[1,5], Hedan Yao[1,5], Ruinan Di[2,5], Zhicheng Xiang[1], Qiang Wang ✉[2 ✉], Fangfang Lu[1], Yu Li[1], Guangxing Yang ✉[3 ✉], Qiang Ma ✉[4] & Zhiping Zhang ✉[1 ✉]

Catalytic reduction of $CO_2$ over Cu-based catalysts can produce various carbon-based products such as the critical intermediate CO, yet significant challenges remain in shedding light on the underlying mechanisms. Here, we develop a modified triple-stage quadrupole mass spectrometer to monitor the reduction of $CO_2$ to CO in the gas phase online. Our experimental observations reveal that the coordinated $H_2O$ on Cu(I)-based catalysts promotes $CO_2$ adsorption and reduction to CO, and the resulting efficiencies are two orders of magnitude higher than those without $H_2O$. Isotope-labeling studies render compelling evidence that the O atom in produced CO originates from the coordinated $H_2O$ on catalysts, rather than $CO_2$ itself. Combining experimental observations and computational calculations with density functional theory, we propose a detailed reaction mechanism of $CO_2$ reduction to CO over Cu(I)-based catalysts with coordinated $H_2O$. This study offers an effective method to reveal the vital roles of $H_2O$ in promoting metal catalysts to $CO_2$ reduction.

[1] School of Chemistry and Chemical Engineering, Xi'an Shiyou University, Xi'an 710065, China. [2] School of Chemistry and Molecular Engineering, Nanjing Tech University, Nanjing 211816, China. [3] School of Chemistry and Chemical Engineering, South China University of Technology, Guangzhou 510641, China. [4] Chinese Academy of Inspection and Quarantine, Beijing 100176, China. [5]These authors contributed equally: Yajun Zheng, Hedan Yao, Ruinan Di. ✉email: wangqiang@njtech.edu.cn; yanggx@scut.edu.cn; zhipingzhang@xsyu.edu.cn

C atalytic reduction of $CO_2$ into high value-added carbon-based products is a promising strategy for tackling current energy demands and reducing greenhouse gas emissions[1-5]. In the past few years, tremendous efforts have been made to explore $CO_2$ reduction reaction ($CO_2$RR), and several products, including $CH_4$[6], $CO$[7], $CH_3OH$[8], $HCOOH$[9], $HCHO$[10], $C_2H_4$[11], $C_2H_6$[12], $C_2H_5OH$[13], and $H_2C_2O_4$[14], have been generated from $CO_2$ reduction via photo-, electro-, or thermal activation[3-5,15-18]. To enhance the selectivity and the conversion efficiency of $CO_2$RR, focus has primarily been on exploring novel catalysts[2,3,19,20]. Despite the progress, many details of the $CO_2$RR mechanisms on the surface of catalysts remain elusive. The techniques used to investigate $CO_2$RR mechanisms include Raman spectroscopy[21], X-ray absorption spectroscopy[22], X-ray photoelectron spectroscopy[23], electron microscopy[24], and calculations using density functional theory (DFT)[25,26]. However, direct observation of the highly reactive intermediates is still a grand challenge.

Mass spectrometry (MS) is a formidable tool for chemical analysis and has also been used to explore the mechanisms of various chemical reactions[27-31]. Nevertheless, there have been few reports on the use of MS to study $CO_2$RR mechanisms in operando[32]. This could be attributed to the following obstacles: (i) $CO_2$ and its resulting products (e.g., CO, $CH_4$, and $CH_3OH$) are neutral molecules and rarely observed in mass spectra without adding charges through ionization; (ii) the lifetimes of reactive intermediates are typically less than milliseconds, making it challenging to capture them with off-line MS. To address these issues, this study proposes a strategy to explore $CO_2$RR mechanisms by adopting the features of triple-stage quadrupole (TSQ) mass spectrometer, in which different stages of quadrupoles in TSQ are used for separation of metal ions and related catalysts originating from nanoelectrospray ionization (nanoESI)[33] (first stage), reaction unit of metal catalysts and $CO_2$ (second stage), and transmission of resulting ions (third stage) (Fig. 1). In the second stage, $CO_2$ and its resulting products could be charged favorably by interacting with metal ions and forming metal complexes. More importantly, the $CO_2$RR processes can be simultaneously carried out with online detection. As such, the reaction pathways of $CO_2$RR can be well identified.

To demonstrate the proof-of-concept, this study explores the reduction reaction of $CO_2$ to CO using copper (Cu) as the catalyst. Cu is the only metal catalyst known to generate hydrocarbons through $CO_2$RR[5,34-39], but poor selectivity and reaction efficiency for value-added products significantly limit its commercial applications[6,40]. Although much effort has been made to understand the catalytic performance of Cu in $CO_2$RR, available techniques that can clarify the underlying reaction mechanisms remain limited[41]. In the $CO_2$RR, CO has been widely identified as a significant product or a primary reaction intermediate for the formation of hydrocarbons[15-17,42,43]. Therefore, understanding the reduction reaction of $CO_2$ to CO over Cu-based catalysts is necessary to promote conversion efficiency.

According to general understanding, there are three routes in $CO_2$ reduction to CO: (i) $CO_2 + H_2 \rightarrow CO + H_2O$[1,17,44], (ii) $*CO_2 + 2e^- + H^+ \rightarrow CO + OH^-$[45], and (iii) $*CO_2 + 2e^- + 2H^+ \rightarrow CO + H_2O$[43,46-50] (Fig. 1). For all three pathways, the O atom in the generated CO originated from $CO_2$. In contrast to the above routes, herein we discovered for the first time that the source of the O atom in CO originated from the $H_2O$ coordinated on transition metal-based catalysts (e.g., Cu, Ag, and Pd) rather than $CO_2$. $H_2O$ is ubiquitous in nature and has been shown to be vital in many reactions such as alcohol oxidation to aldehyde[51], $CH_4$ oxidation to $CH_3OH$[52,53], and $CO_2$ reduction to $CH_4$, $CH_3OH$, and $HCOOH$[54]. However, the role of $H_2O$ in the reduction of $CO_2$ to CO is ambiguous. To elucidate the role of $H_2O$ in $CO_2$RR, we investigated the behavior of $Cu^+$ and $[Cu(H_2O)]^+$ and found that the coordinated $H_2O$ in $[Cu(H_2O)]^+$ not only favored the adsorption of $CO_2$ onto $Cu^+$, but also facilitated the reduction of $CO_2$ to CO. Isotope-labeling studies provided evidence suggesting that the origin of the O atom in CO was from the coordinated $H_2O$ on $Cu^+$. By combining experimental results and computational calculations with DFT, the detailed reaction mechanism of $CO_2$ reduction to CO over a $[Cu(H_2O)]^+$ catalyst was proposed. The data presented in this work allowed us to elucidate the role of $H_2O$ in $CO_2$RR and offered new insights for developing effective systems that enhanced the selectivity and conversion efficiency of $CO_2$ to CO and other carbon-based products.

## Results and Discussion

**Reaction apparatus and generation of Cu(I) species.** To reveal the underlying roles of $H_2O$ in the reduction of $CO_2$ to CO via copper-based catalysis, we employed a modified TSQ apparatus for online observation of the $CO_2$RR and detection of the reaction intermediates and products. As shown in Fig. 1, nanoESI was used as an ionization source to generate Cu-based ions. As the catalyst ions were introduced into the TSQ apparatus, the desired Cu species were isolated from Q1 and then transferred to Q2. In the reaction unit of Q2, Cu-based ions interacted with $CO_2$ upon the applied voltage of 5 V and formed CO. Due to the low reaction efficiency at low temperatures (as discussed below), $CO_2$ gas was heated to 280 °C using a heating tape in the gas circuit system prior to reacting with the Cu catalyst in the gas phase. After the reduction reaction was completed, the resulting products were transferred directly to Q3 followed by online detection.

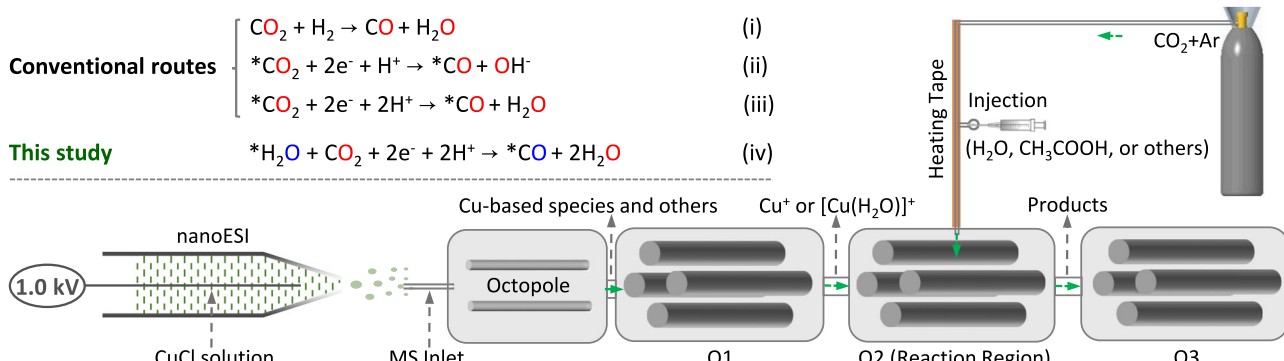

**Fig. 1 Schematic diagram of the apparatus for $CO_2$ reduction and detection of reaction products.** (insets in the top left corner are the different routes for generation of CO from $CO_2$, in which * means catalyst).

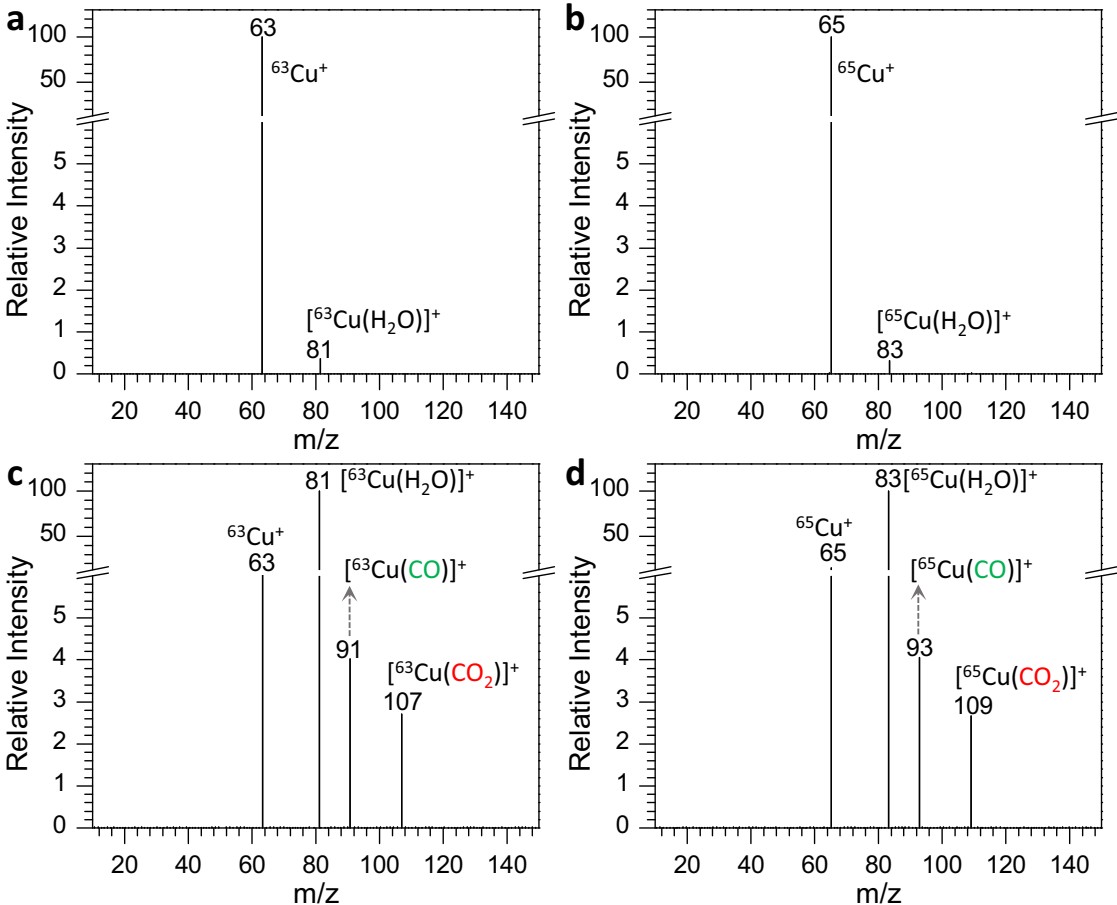

**Fig. 2 Mass spectra of $CO_2$ reduction to CO under different Cu-based catalytic systems. a** $^{63}Cu^+$ **b** $^{65}Cu^+$. **c** $[^{63}Cu(H_2O)]^+$. **d** $[^{65}Cu(H_2O)]^+$ (gas circuit temperature: 280 °C; reaction pressure: 1.5 mTorr).

Many previous studies[8,26,55–59] have demonstrated that of the different oxidation states, Cu(I)-related species are one of the most important catalysts in CO₂RR. Considering that the current study investigated different Cu-based complexes in the reduction of $CO_2$, nanoESI was utilized to generate Cu(I) species rather than inductively coupled plasma (ICP)[60,61], which only produces Cu(I). In the nanoESI process, the types of Cu species and solvent were found to have a pronounced effect on the resulting mass spectra. When CuX (X = Cl, Br, or I) was dissolved into acetonitrile, comparable and intensive peaks of Cu-based ions appeared in the mass spectra (e.g., $^{63}Cu^+$, $^{65}Cu^+$, $[^{63}Cu(H_2O)]^+$, and $[^{65}Cu(H_2O)]^+$), and few non-Cu species emerged (Supplementary Figs. 1–3). Thus, CuCl was used to generate different Cu(I)-based ions.

**Effect of $H_2O$ on the reduction of $CO_2$ to CO**. The above MS setup allowed us to investigate the reduction of $CO_2$ to CO in situ. As $^{63}Cu^+$ or $^{65}Cu^+$ ions were isolated to interact with $CO_2$, no obvious resultant species were observed, and only $Cu^+$ itself and $[Cu(H_2O)]^+$ appeared in the mass spectra (Fig. 2a, b). Remarkably, when $[^{63}Cu(H_2O)]^+$ or $[^{65}Cu(H_2O)]^+$ was introduced to the reaction unit along with $CO_2$, not only did the reaction product, CO, emerge in the forms of $[^{63}Cu(CO)]^+$ and $[^{65}Cu(CO)]^+$, but so did the reactant, $CO_2$, as complexes of $[^{63}Cu(CO_2)]^+$ and $[^{65}Cu(CO_2)]^+$, in addition to $[^{63}Cu(H_2O)]^+$, $[^{65}Cu(H_2O)]^+$, and their fragment ions $^{63}Cu^+$ and $^{65}Cu^+$ (Fig. 2c, d). These results suggested that in contrast to the bare $Cu^+$, the coordinated $H_2O$ in $[Cu(H_2O)]^+$ played a vital role in the interaction with $CO_2$, which both rendered the adsorption of

$CO_2$ and the reduction of $CO_2$ to CO on the Cu(I) catalyst. We also found that the coordinated $H_2O$ favored other metal-based ions (e.g., $[Ag(H_2O)]^+$ and $[Pd(H_2O)]^+$) to carry out the CO₂RR (Supplementary Figs. 4 and 5), revealing that $H_2O$ molecules on the metal catalysts were active sites for facilitating the effective reduction of $CO_2$ to CO. The effect of $H_2O$ could also be mirrored by a reversed water-gas shift reaction (WGSR) using Cu/ $ZnO/Al_2O_3$ as catalyst (Supplementary Fig. 6a–c) and in situ diffuse reflectance infrared Fourier transform spectroscopy (in situ DRIFTS) using Cu/γ-$Al_2O_3$ and Pt/γ-$Al_2O_3$ as catalysts (Supplementary Fig. 7). It hinted that dosing a suitable amount of $H_2O$ could promote heterogeneous thermal catalytic conversion of $CO_2$ to CO in realistic conditions.

To quantitatively describe the roles of coordinated $H_2O$ over metal surfaces to $CO_2$ adsorption and reduction to CO, we used Cu-based catalysts as an example. Fig. 3a and Supplementary Fig. 8a show the effect of reaction gas pressure on the adsorption of $CO_2$ to $Cu^+$. The adsorption performance was examined by the absolute peak intensities of generated $[Cu(CO_2)]^+$ in mass spectrometric analysis. When bare $Cu^+$ was employed, the adsorption ability of $CO_2$ onto the $Cu^+$ catalyst gradually increased with the increasing reaction gas pressure from 0 to 4 mTorr. However, in the presence of $[Cu(H_2O)]^+$, the adsorption of $CO_2$ onto $Cu^+$ increased with increasing reaction gas pressure and reached a maximum value in the range of 1.5 – 2.5 mTorr, then decreased thereafter. More interestingly, in the presence of $[^{63}Cu(H_2O)]^+$, the peak intensity of $[^{63}Cu(CO_2)]^+$ was 48.6-fold higher than that with $^{63}Cu^+$ in their optimal conditions. This demonstrated that after $H_2O$ was coordinated to

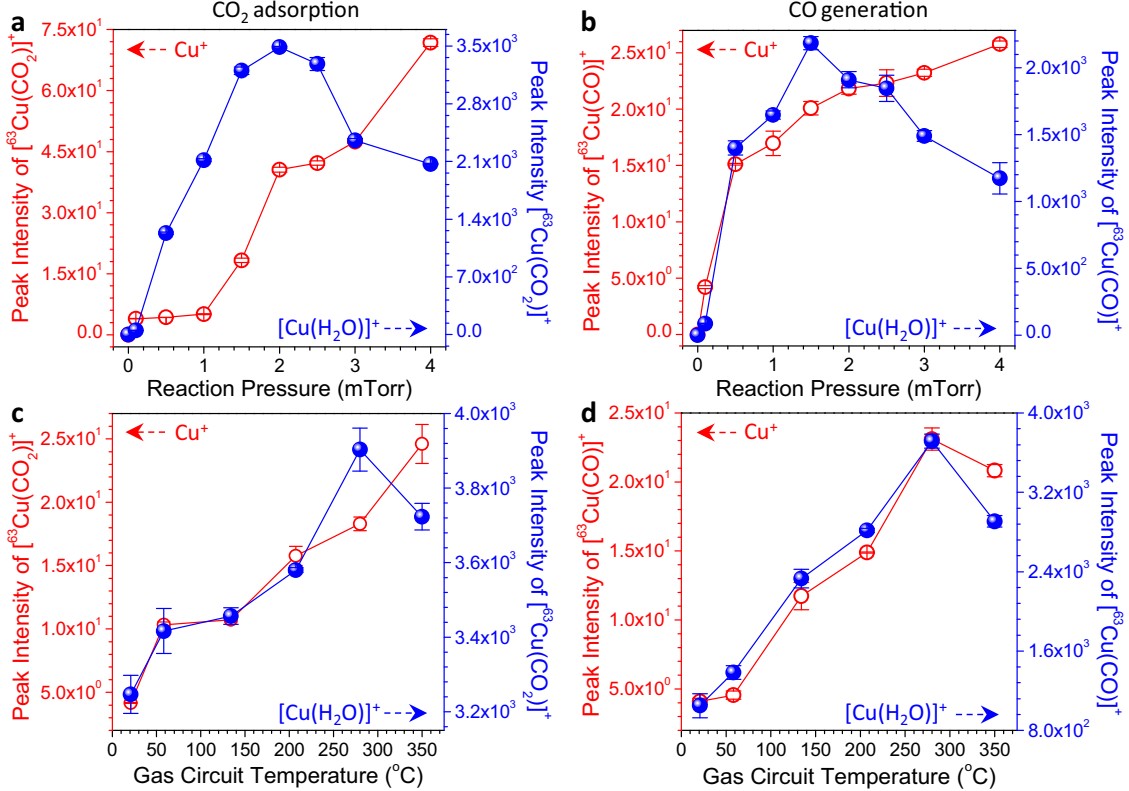

**Fig. 3 Variation of CO₂ adsorption and CO generation with reaction pressure and temperature of heating tape around the gas circuit.** Effect of reaction pressure on (**a**) the adsorption of $CO_2$ onto $^{63}Cu^+$ and (**b**) CO generation by $CO_2$ reduction in the presence of either $^{63}Cu^+$ or $[^{63}Cu(H_2O)]^+$ catalysts (gas circuit temperature: 280 °C). Effect of the gas circuit temperature on (**c**) the adsorption of $CO_2$ onto $Cu^+$ and (**d**) CO generation by $CO_2$ reduction in the presence of either $^{63}Cu^+$ or $[^{63}Cu(H_2O)]^+$ catalysts (reaction pressure: 1.5 mTorr).

$Cu^+$, $CO_2$ was more prone to interact with $[^{63}Cu(H_2O)]^+$ to generate $[^{63}Cu(CO_2)]^+$ compared to bare $Cu^+$. Besides $CO_2$, we also found that in contrast to $Cu^+$, $[Cu(H_2O)]^+$ was likely to form Cu-based complexes with other molecules such as methanol (Supplementary Fig. 9), ethanol (Supplementary Fig. 10), acetonitrile (Supplementary Fig. 11), benzene (Supplementary Fig. 12), toluene (Supplementary Fig. 13), and dichloromethane (Supplementary Fig. 14). Such a fact suggests that the coordinated $H_2O$ in the structure of $[Cu(H_2O)]^+$ is a uniquely active site for adsorbing different molecules. To our knowledge, it could be speculated as the following reasons. As $H_2O$ was bound to $Cu^+$, the resulting $[Cu(H_2O)]^+$ more likely tended to form hydrogen bond[62–66] or OH-π interactions[67–69] with those studied molecules than bare $Cu^+$. After undergoing further structural rearrangements, Cu(I)-based complexes were favorably generated. Despite this, detailed reasons need to be further studied.

It is well known that adsorption onto catalyst surfaces is a very crucial step for $CO_2$ reduction, and a more favorable $CO_2$ adsorption can facilitate a more efficient reduction to CO[15]. To evaluate the reduction ability of $CO_2$ to CO in the presence of $Cu^+$ and $[Cu(H_2O)]^+$, we compared the difference in quantity of generated $[Cu(CO)]^+$ at varying gas pressures (Fig. 3b and Supplementary Fig. 8b). For bare $Cu^+$, the peak intensity of $[^{63}Cu(CO)]^+$ sharply increased at 0 – 0.5 mTorr, followed by a gradual increase at 1 – 4 mTorr. A different pattern was observed for $[Cu(H_2O)]^+$, where the peak intensity first increased to a maximum value at 1.5 mTorr, and then decreased thereafter. Comparing both systems with $Cu^+$ and $[Cu(H_2O)]^+$, it was remarkable that $[Cu(CO)]^+$ was generated more favorably for $[Cu(H_2O)]^+$. Similar to the amount of adsorbed $CO_2$, the amount of CO generated in the presence of $[^{63}Cu(H_2O)]^+$ was 84.7-fold

higher than that in the presence of $^{63}Cu^+$ under optimal conditions. These results indicated that the coordinated $H_2O$ in $[^{63}Cu(H_2O)]^+$ not only greatly facilitated $CO_2$ adsorption to $Cu^+$, but also significantly promoted CO generation.

A similar process was also conducted to compare the adsorption of $CO_2$ and generation of CO at different temperatures, which was controlled by varying the temperature of heating tape around the gas circuit (Fig. 1). As the temperature increased from room temperature (20 °C) to 350 °C, $CO_2$ adsorption using $^{63}Cu^+$ demonstrated a gradual increasing pattern. However, a different pattern was observed for the system using $[^{63}Cu(H_2O)]^+$, in which an initial steady increasing trend from room temperature to 280 °C was observed, followed by a slight decreasing one within the range of 280 – 350 °C (Fig. 3c and Supplementary Fig. 8c). In the optimal temperature range, the $CO_2$ adsorption amount in the presence of $[^{63}Cu(H_2O)]^+$ was more than two orders of magnitude (158.7-fold) higher than $^{63}Cu^+$. For the generation of CO, both $^{63}Cu^+$ and $[^{63}Cu(H_2O)]^+$ catalytic systems demonstrated a comparable pattern, in which the formation of CO first increased and reached a maximum value at 280 °C, then decreased thereafter (Fig. 3d and Supplementary Fig. 8d). By comparing both systems, it was found that the use of $[^{63}Cu(H_2O)]^+$ resulted in more than two orders of magnitude (160.9-fold) of CO formation than using $^{63}Cu^+$. Thus, we can conclude in confident that the coordinated $H_2O$ in $[^{63}Cu(H_2O)]^+$ was a governing factor in $CO_2$ adsorption and CO generation.

From the above, it was noticeable that there was a maximum $CO_2$ adsorption capacity and a maximum CO production rate when $[Cu(H_2O)]^+$ was used as catalyst throughout the studied temperature and pressure ranges (Fig. 3). This phenomenon was presumably due to the instability of $[Cu(H_2O)]^+$ with increasing

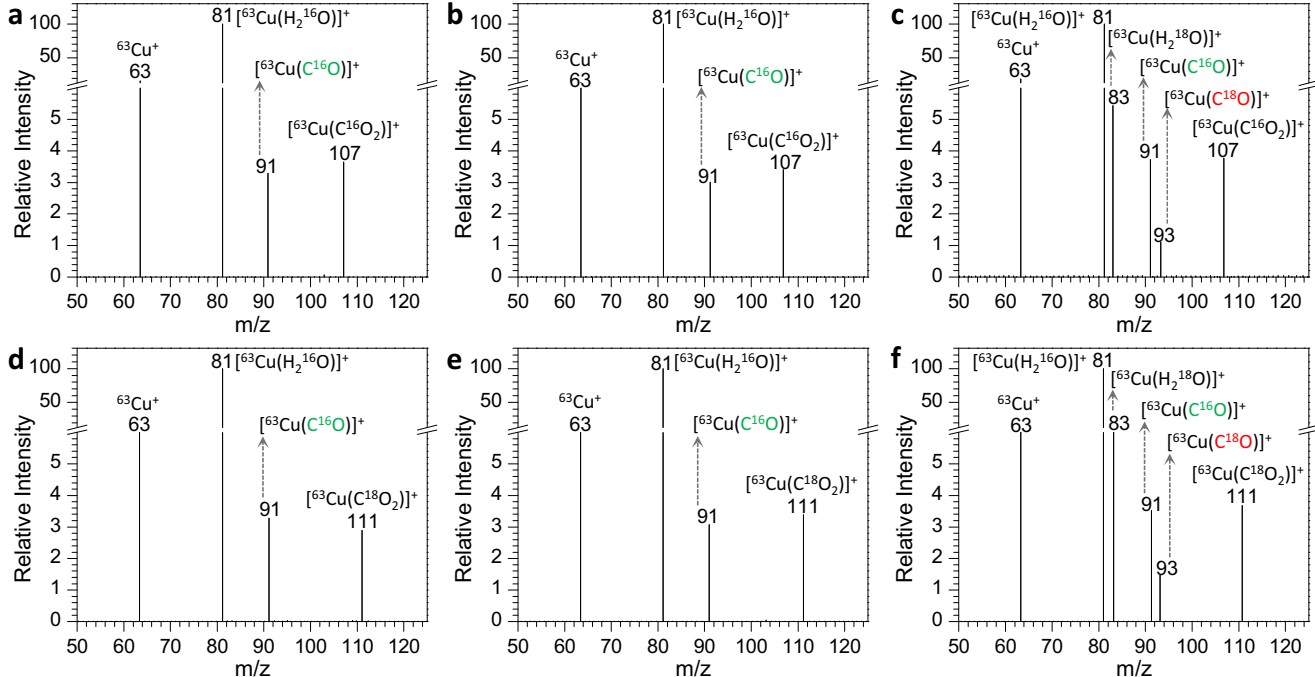

**Fig. 4 Isotope-labeling MS measurement results under different systems. a** $[^{63}Cu(H_2^{16}O)]^+$ and $C^{16}O_2$. **b** $[^{63}Cu(H_2^{16}O)]^+$, $H_2^{16}O$, and $C^{16}O_2$. **c** $[^{63}Cu(H_2^{16}O)]^+$, $H_2^{18}O$, and $C^{16}O_2$. **d** $[^{63}Cu(H_2^{16}O)]^+$ and $C^{18}O_2$. **e** $[^{63}Cu(H_2^{16}O)]^+$. $H_2^{16}O$, and $C^{18}O_2$; **f** $[^{63}Cu(H_2^{16}O)]^+$, $H_2^{18}O$ and $C^{18}O_2$ (gas circuit temperature: 280 ºC; reaction pressure: 1.5 mTorr).

reaction pressures and temperatures. As an example of increasing reaction pressures, the content of $CO_2$ in the Q2 of mass spectrometer (Fig. 1) steadily increased and therefore, high yields of $[Cu(CO_2)]^+$ and $[Cu(CO)]^+$ should be generated, whereas an opposite trend was observed within the range of 2 – 4 mTorr (Fig. 3a, b). As aforementioned, $[Cu(H_2O)]^+$ was more favorable for the generation of $[Cu(CO_2)]^+$ and $[Cu(CO)]^+$ than $Cu^+$. However, the generated amount of $[Cu(H_2O)]^+$ demonstrated a decreasing pattern with increasing reaction pressures from 2 to 4 mTorr (Supplementary Fig. 15a). A lower amount of $[Cu(H_2O)]^+$ would result in a lower generation efficiency to $[Cu(CO_2)]^+$ and $[Cu(CO)]^+$. As a compromise between reaction pressure and the amount of $[Cu(H_2O)]^+$, 1.5 mTorr gave the optimal performance. As to the temperature, the same pattern as the reaction pressure was observed (Fig. 3c, d), and 280 °C offered the highest generation efficiencies to both $[Cu(CO_2)]^+$ and $[Cu(CO)]^+$. This case may be associated with the compromise of $[Cu(H_2O)]^+$ stability (Supplementary Fig. 15b) and the thermodynamic reaction activity between $CO_2$ and $[Cu(H_2O)]^+$ under increasing temperatures.

**Origin of the O atom in generated CO**. Isotope-labeling experiments ($H_2^{16}O$, $H_2^{18}O$, $C^{16}O_2$, and $C^{18}O_2$) allowed us to conclude that the origin of the O atom in the generated CO was from the coordinated $H_2O$, rather than $CO_2$ itself. In previous studies[1,17,34,43–50], different mechanistic scenarios were proposed to elucidate the reduction pathway of $CO_2$ to CO, and the generation of CO was conventionally contributed to the loss of an O atom in $CO_2$. However, no direct experimental evidence generated through isotopic labeling has been supported thus so far. MS has been demonstrated to be a very powerful tool that offers the opportunity to derive the origin of the O atom in generated CO from $CO_2$.

As expected, monitoring of the reaction between $[^{63}Cu(H_2^{16}O)]^+$ and $C^{16}O_2$ resulted in the observation of $[^{63}Cu(C^{16}O)]^+$ (Fig. 4a). The inability to isolate $[^{63}Cu(H_2^{18}O)]^+$, by dissolving CuCl or other Cu-based chemicals into the mixture of acetonitrile and $H_2^{18}O$,

meant that it was impossible to probe the reaction products in the presence of $[^{63}Cu(H_2^{18}O)]^+$ and $C^{16}O_2$. This case could be attributable to the fact that there are plenty of $H_2^{16}O$ in air, and the $H_2^{18}O$ in generated $[^{63}Cu(H_2^{18}O)]^+$ would be quickly exchanged by $H_2^{16}O$ in the plume of nanoESI or the transfer process to the Q1 of mass spectrometer (Fig. 1). As a result, insufficient $[^{63}Cu(H_2^{18}O)]^+$ ions could be generated and were not favorable for MS detection. To resolve this issue, $[^{63}Cu(H_2^{16}O)]^+$ was first isolated in Q1, and $H_2^{16}O$ or $H_2^{18}O$ was then injected into the $CO_2$ gas circuit (Fig. 1). After interaction among the reactants in the reaction region Q2, the resulting products were analyzed. As $H_2^{16}O$ was injected (Fig. 4b), the mass spectrum was analogous to that of only $[^{63}Cu(H_2^{16}O)]^+$ and $C^{16}O_2$ (Fig. 4a). Noticeably, when $H_2^{18}O$ was added to the system containing $[^{63}Cu(H_2^{16}O)]^+$ and $C^{16}O_2$ (Fig. 4c), not only did $[^{63}Cu(H_2^{18}O)]^+$ appear in the mass spectrum, but so did $[^{63}Cu(C^{16}O)]^+$ and $[^{63}Cu(C^{18}O)]^+$. These results indicated that it was an effective strategy for generating $[^{63}Cu(H_2^{18}O)]^+$, and more importantly, the origin of the O atom in CO was presumable from the coordinated water.

More convincing evidence was gained from the reaction between $[^{63}Cu(H_2^{16}O)]^+$, $C^{18}O_2$, and $H_2^{16}O/H_2^{18}O$. Interestingly, as $[^{63}Cu(H_2^{16}O)]^+$ interacted with $C^{18}O_2$ (Fig. 4d), $[^{63}Cu(C^{16}O)]^+$, rather than $[^{63}Cu(C^{18}O)]^+$, was formed, indicating that the O atom in CO was indeed from the $H_2O$. The same result was obtained from the system with $[^{63}Cu(H_2^{16}O)]^+$, $C^{18}O_2$, and $H_2^{16}O$ (Fig. 4e). However, as $H_2^{18}O$ was injected into the $C^{18}O_2$ gas circuit system (Fig. 4f), the mass spectrum was analogous to the system with $[^{63}Cu(H_2^{16}O)]^+$, $C^{16}O_2$ and $H_2^{18}O$ (Fig. 4c), in which both $[^{63}Cu(C^{16}O)]^+$ and $[^{63}Cu(C^{18}O)]^+$ were generated. This phenomenon, as well as $^{65}Cu$ (Supplementary Fig. 16), $^{107}Ag$, and $^{104}Pd$ (Supplementary Fig. 17)-based reactions, allowed us to draw a definite conclusion that in the $CO_2RR$ process, the two O atoms in $CO_2$ were eliminated completely by losses of two molecules of $H_2O$[46,48,70], and the O atom in CO originated from the coordinated $H_2O$. Moreover, $H_2^{18}O$-labeling experiments in tandem with off-line GC-MS

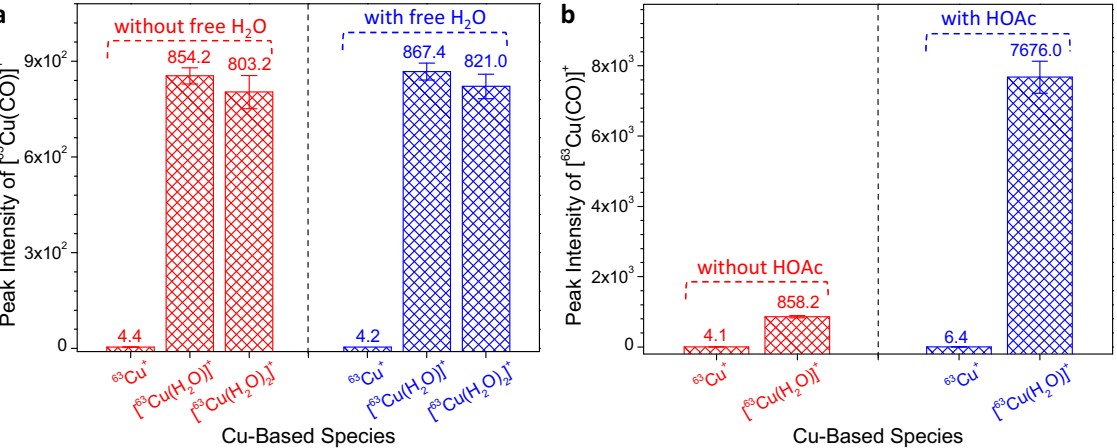

**Fig. 5 Influence of the number of coordinated H₂O, free H₂O, and extraneous acid on the generation of CO. a** Effects of the number of coordinated $H_2O$ and free $H_2O$ on $CO_2$ reduction to CO under different Cu-based catalytic systems ($^{63}Cu^+$, $[^{63}Cu(H_2O)]^+$, and $[^{63}Cu(H_2O)_2]^+$). **b** Effect of extraneous acid on the reduction of $CO_2$ to CO under different Cu-based catalytic systems. Note: The free $H_2O$ and HOAc (acetic acid) were injected into the gas circuit system by an injector; gas circuit temperature: 280 ºC; reaction pressure: 1.5 mTorr; $n = 5$).

analysis were carried out to confirm the source of O atom in resulting CO from the reversed WGSR. As shown in Supplementary Fig. 6f–h, in comparison with the system of introducing $H_2^{16}O$, an abundant peak of m/z 30 ($C^{18}O$) emerged in the mass spectrum for the system of $H_2^{18}O$. This fact indicated that the involved $H_2O$ was the oxygen source of the resulting CO. More importantly, the observed results of $CO_2$ reduction in the TSQ mass spectrometer could be extrapolated to realistic heterogeneous catalysis. Those gas-phase results also correlated directly to solution-phase $CO_2$RR. To gain insight into this point, we performed the electrochemical reduction of $CO_2$ in KCl aqueous solution using Au electrode, because the produced amount of CO was below the limit of detection using Cu or Ag electrode. The reaction products were monitored using an online differential electrochemical mass spectrometer (Supplementary Fig. 18). In contrast to the system of $H_2^{16}O$, a considerable amount of $C^{18}O$ (m/z 30) was generated in the presence of $H_2^{18}O$ (Supplementary Figs. 19–21). This further indicated that $H_2O$ was the oxygen source of resulting CO from $CO_2$ reduction and provided solid evidence on the generalization of the current finding to related reactions in the condensed phase (Supplementary Table 1).

**Identifying the effect of H₂O on CO generation.** The number of coordinated $H_2O$ in $[^{63}Cu(H_2O)_x]^+$ ($x = 0$, 1, or 2) was observed to have a significant influence on the reduction of $CO_2$ to CO. The reaction efficiency was evaluated by comparing the absolute peak intensities of generated $[^{63}Cu(CO)]^+$ in mass spectrometric analysis. When bare $^{63}Cu^+$ was used as the catalyst, a low signal was observed (Fig. 5a and Supplementary Fig. 22). Noticeably, a significant improvement was observed for $[^{63}Cu(H_2O)]^+$, and the reaction efficiency was 193.8 times higher than with $^{63}Cu^+$. However, further increasing the number of coordinated $H_2O$ to 2 (e.g., $[^{63}Cu(H_2O)_2]^+$) resulted in a comparable reaction efficiency to $[^{63}Cu(H_2O)]^+$, indicating that one coordinated $H_2O$ molecule was sufficient for high-efficiency reduction of $CO_2$ to CO.

In the reaction between $[^{63}Cu(H_2O)]^+$ and $CO_2$, it was unclear how $H_2O$ molecules interacted with $CO_2$, namely whether it was a (i) direct interaction between $[^{63}Cu(H_2O)]^+$ and $CO_2$, or if (ii) $CO_2$ first interacted with the dissociated $H_2O$ molecule from $[^{63}Cu(H_2O)]^+$, then reacted with dissociated $^{63}Cu^+$ or $[^{63}Cu(H_2O)]^+$. To probe the above assumptions, 2 μL of free $H_2O$ was injected into the $CO_2$ gas circuit system at a temperature of 280 °C (Fig. 1). After being injected, the free $H_2O$ molecules

immediately evaporated and interacted with $CO_2$ in the gas circuit by forming $H_2CO_3$ ($H_2O + CO_2 \rightarrow H_2CO_3$)[71–73], which was indirectly confirmed by the isotope-labeling experiments, namely $H_2^{18}O$ and $C^{16}O_2$ (Supplementary Fig. 23). If route (ii) was taken in the $CO_2$RR, the involved free $H_2O$ would facilitate $CO_2$ reduction to CO. However, a comparable reaction efficiency to the system without free $H_2O$ was observed (Fig. 5a and Supplementary Fig. 22), indicating that $CO_2$ reduction to CO occurred through route (i). Namely, $[Cu(H_2O)]^+$ first interacted with $CO_2$ by a formation of $[Cu(H_2O)(CO_2)]^+$, which was supported by isotope-labeling experiments (Supplementary Fig. 24), and then a reduction reaction occurred to convert $CO_2$ to CO. The above results also indicated that the coordinated $H_2O$ on the $Cu^+$, rather than free $H_2O$, played crucial roles for the reduction of $CO_2$ to CO. The transition metals such as $Cu^+$ not only provided efficient active sites for the generation of $[Cu(H_2O)]^+$ or other $H_2O$-based metal complex ions, but also offered opportunity to charge neutral species such as $CO_2$ and CO for favorable mass spectrometric analysis by the forms of $[Cu(CO_2)]^+$ and $[Cu(CO)]^+$ (Figs. 2 and 4).

Based on the prior experiments, the origin of the O atom in CO was from $H_2O$ rather than $CO_2$. Thus, the two O atoms in $CO_2$ needed to be eliminated prior to the formation of CO. According to the previous $CO_2$RR mechanisms[46,48,70], the generation of CO was achieved by first proton-electron ($H^+/e^-$) transfer processes to $CO_2$, followed by the elimination of $H_2O$. To promote the formation of CO, an efficient step to proton generation and capture was necessary[46]. In this study, a proton ($H^+$) was also found to be crucial to the generation of CO. For the reaction of $[Cu(H_2O)]^+$ and $CO_2$, the generation of $H^+$ was believed to be caused by water dissociation. However, further dissociation of water might affect its ability to participate in the formation of $[Cu(H_2O)(CO_2)]^+$ and reduce the reaction efficiency. Addition of supplementary $H^+$ could mitigate water dissociation and boost CO generation. To confirm this hypothesis, 2 μL of acetic acid (HOAc) was injected into the $CO_2$ gas circuit system. For the system containing $[Cu(H_2O)]^+$, a 8.9-fold increase of CO formation was observed with the addition of HOAc, whereas no significant difference was observed for bare $Cu^+$ (Fig. 5b and Supplementary Figs. 25 and 26). These results indicated that supplementary $H^+$ was indeed important for boosting the reduction efficiency of $CO_2$ to CO in the presence of $[Cu(H_2O)]^+$ catalyst, in which $H^+$ from the coordinated $H_2O$ participated into the elimination of O atoms in $CO_2$ [46,48,70].

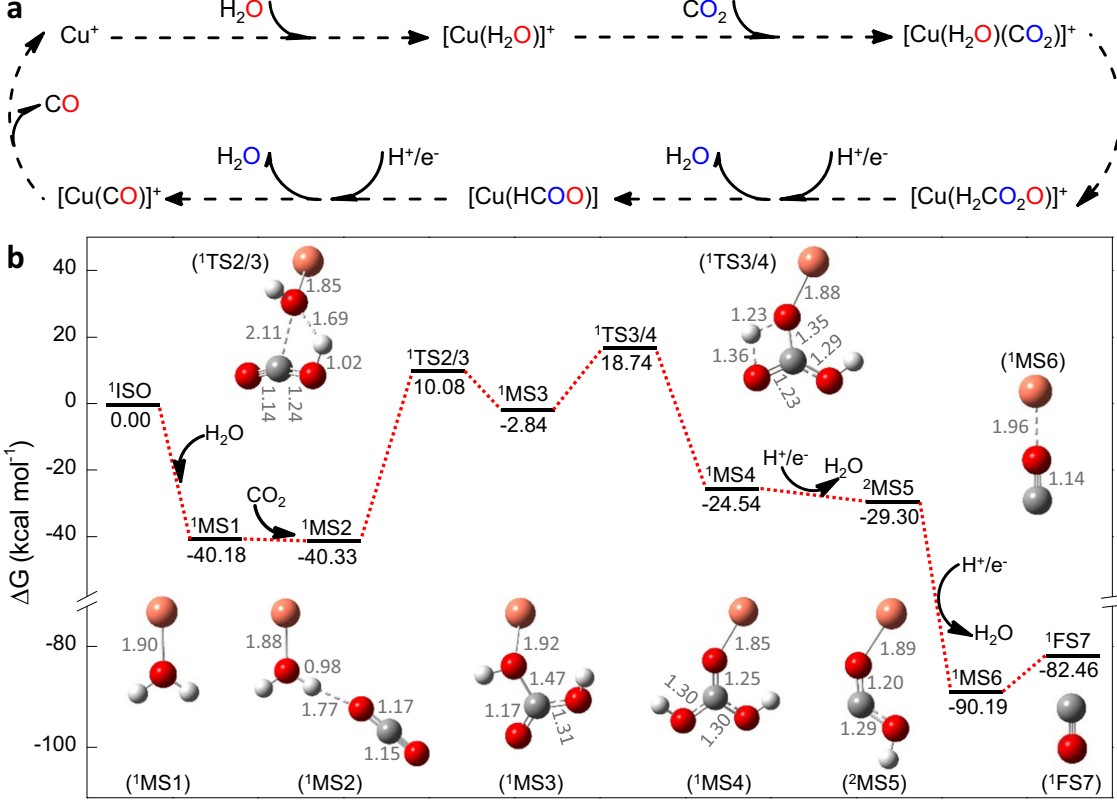

**Fig. 6 The mechanism of $CO_2$ reduction to CO in the presence of Cu(I) catalyst and water. a** Schematic representation of the reduction of $CO_2$ to CO according to experimental observations. **b** The DFT microscopic reaction pathway [the singlet reactants of $^1Cu^+ + ^1H_2O + ^1CO_2$ ($^1$ISO, 0.00) was taken to be zero as a reference], demonstrating the thermodynamic and kinetic feasibility of the suggested pathway.

Many previous studies[8,26,55–59,74,75] illustrated that the Cu(I) surface was the active site anchoring the $CO_2$RR. Because of this, several strategies, including the addition of oxygen[59] or copper nitride support[76], boron-element doping[39], plasma treatment[77], catalyst electro-redeposition[78], and covering the catalyst surface with nanocavities[55], have been used to stabilize the active Cu(I) oxidation state. Despite these efforts, the precise effect of a single Cu valence state on the reduction of $CO_2$ to CO remains ambiguous. Ions can be easily isolated through MS, enabling the investigation of the interaction between $CO_2$ and Cu with different oxidation states. In this study, we not only investigated the effect of Cu(I) in $CO_2$RR, but also explored the performance of Cu(II) on CO production. Taking the copper oxidation state and the aforementioned $H_2O$ effect into account, $[^{63}Cu(OH)\cdot H_2O]^+$ and $[^{65}Cu(OH)\cdot H_2O]^+$ (Supplementary Fig. 27a, b) were isolated to interact with $CO_2$. However, in contrast to the Cu(I) system (Fig. 4), a much lower amount of $[^{63}Cu(CO)]^+$ or $[^{65}Cu(CO)]^+$ was observed in the mass spectrum (Supplementary Fig. 27c, d). These results indicated that Cu(I) was better at reducing $CO_2$ to CO than Cu(II) even in the presence of water, which was in good agreement with the above discussion.

**Mechanistic studies**. Based on the overwhelming evidence from the experimental observations, the pathway of $CO_2$ reduction to CO over Cu(I)-based catalyst was suggested in Fig. 6a. This process started with the formation of $[Cu(H_2O)]^+$, which interacted with $CO_2$ to generate $[Cu(H_2O)(CO_2)]^+$. As afore-mentioned, $H_2CO_3$ formed after the reaction of $CO_2$ and $H_2O$, thereby the occurrence of transition from $[Cu(H_2O)(CO_2)]^+$ to $[Cu(H_2CO_3)]^+$, which was similar to the direct $CO_2$ capture and conversion to fuels over magnesium nanoparticles[54]. Sub-sequent $H^+/e^-$ transfer reactions[46,48,70] led to the generation

of $[Cu(CO)]^+$, along with the elimination of two $H_2O$ molecules. In the $CO_2$RR, $H^+$ ions originated from the dissociation of $H_2O/H_2CO_3$ or added acid, and the necessary electrons involved in the reduction of $CO_2$ to CO were likely from the generated $H_2CO_3$ (Supplementary Fig. 28 and related discussion). Upon the $CO_2$ reduction, the O atom in $[Cu(H_2O)]^+$ combined with the C atom in $CO_2$ by the formation of CO, whereas the two O atoms in $CO_2$ were eliminated by loss of two $H_2O$ molecules. The released $H_2O$ molecules could combine with $Cu^+$ to form $[Cu(H_2O)]^+$ for the next cycle of $CO_2$ reduction. Despite this, the oxidation state of copper persisted +1, in good agreement with the previous report[77].

To gain detailed insight into the reaction mechanism, DFT calculations were carried out to explore the key reaction intermediates and possible reaction pathways. Both the singlet and triplet reaction paths had been considered to understand all possibilities. The singlet reaction path was more thermodynami-cally favorable than the triplet reaction path, and both paths were similar (Fig. 6b, Supplementary Fig. 29, and Supplementary Table 2–4). Because of this fact, the discussion herein focused mainly on the singlet reaction path with a lower energy.

The activation of $CO_2$ was initialized with the collision interaction among $^1CO_2$, $^1Cu^+$, and $^1H_2O$ (Supplementary Fig. 30). All calculations on bimolecular interactions indicated that $^1[Cu-OH_2]^+$ ($^1$MS1), through the collision of $^1Cu^+$ and $^1H_2O$, was the most preferred possibility in contrast to others, such as $^1Cu^+$ and $^1CO_2$ or $^1CO_2$ and $^1H_2O$. The binding energy of $^1Cu^+$ and $^1H_2O$ in $^1$MS1 was as high as $-40.18$ kcal mol$^{-1}$, which was larger than those of $^1Cu^+$ and $^1CO_2$ ($^1[Cu-O=C=O]^+$, $-19.68$ kcal mol$^{-1}$) and $^1CO_2$ and $^1H_2O$ ($^1[H_2O-O=C=O]^+$, 2.53 kcal mol$^{-1}$). In the collision between $^1Cu^+$ and $^1H_2O$, $^1Cu^+$ was prone to interact with the central O atom in $H_2O$ to form $^1[Cu-OH_2]^+$ through O-Cu coupling with a bond length of 1.90 Å.

The existence of activated $^1[Cu\text{-}OH_2]^+$ ($^1MS1$) promoted the reduction of $^1CO_2$. First, $^1CO_2$ interacted with $^1MS1$ to form the $^1[Cu\text{-}O(H)\text{-}H\cdots OCO]^+$ ($^1MS2$, $-40.33$ kcal mol$^{-1}$) intermediate by a weak O-H···O hydrogen bond. The O-C bond length of $^1CO_2$ changed from 1.16 Å to 1.17 and 1.15 Å in $^1MS2$. Subsequently, $^1MS2$ isomerized into $^1[Cu\text{-}O(H)\text{-}C(O)OH]^+$ ($^1MS3$, $-2.84$ kcal mol$^{-1}$) through a transition state, $^1TS2/3$ (10.08 kcal mol$^{-1}$). It is worth noting that in the conversion from $^1TS2/3$ to $^1MS3$, the breakage of the O-H bond in $H_2O$ occurred, and the O-H and C-O bonds formed between the $H_2O$ and $CO_2$ (Supplementary Fig. 31–34). This process was the rate-limiting step of $CO_2$ reduction with an activation barrier as high as 50.41 kcal mol$^{-1}$. Further isomerization turned $^1MS3$ into $^1[Cu\text{-}O\text{-}C(OH)_2]^+$ ($^1MS4$, $-2.84$ kcal mol$^{-1}$), which only involved the migration of intramolecular hydrogen through the transition state $^1TS3/4$ (18.74 kcal mol$^{-1}$), and the activation barrier from $^1MS3$ to $^1TS3/4$ was 21.58 kcal mol$^{-1}$.

Once $^1MS4$ was generated, two molecules of water were lost by a H$^+$/e$^-$ transfer process, thereby leading to the formation of $^1[Cu\text{-}O\text{-}C]^+$ ($^1MS6$, $-90.19$ kcal mol$^{-1}$). In the process, H$^+$/e$^-$ was initially transferred from the reaction system to the OH group of $^1MS4$, which resulted in the release of one $H_2O$ molecule to form a $^1[Cu\text{-}O\text{-}COH]^+$ ($^2MS5$, $-29.30$ kcal mol$^{-1}$) intermediate. As the OH group in $^2MS5$ was further attacked by H$^+$/e$^-$, a $^1MS6$ intermediate was formed through the release of another $H_2O$ molecule. The dehydration process would compete with hydrogen evolution reaction (HER). By calculation of their corresponding free energy, the values were $-4.76$ kal mol$^{-1}$ and $-60.89$ kcal mol$^{-1}$ when $^1MS4$ and $^2MS5$ were transferred to $^2MS5$ and $^1MS6$, respectively. For a standard hydrogen electrode, the free energy is $-19.09$ kal mol$^{-1}$ as H$^+$ is transferred to $H_2$ in basic solution, which system is favorable to aqueous electrochemical CO$_2$RR while preventing from HER. Apparently, in contrast to the transfer of $^1MS4$ to $^2MS5$, it was prone to the HER, whereas an opposite trend occurred when $^2MS5$ was transferred to $^1MS6$. Along with the processes, $^2MS5$ would also tend to combine with H$^+$ by formation of $[Cu(HCOOH)]^+$ as a side reaction, which was captured in our current study (Supplementary Figs. 7a, b and 35).

In the last step, the bound CO group dissociated from $^1MS6$ by forming an isolated CO molecule and pristine $^1Cu^+$, thus completing the catalytic cycle and releasing $-82.46$ kcal mol$^{-1}$ of thermal energy. The reaction pathway offers further convincing theoretical evidence for the hypothesized mechanism derived from mass spectrometric analysis. It also details how $H_2O$ molecule is involved in the reduction of $CO_2$ to CO and how it replaces two O atoms in $CO_2$ with the one in its structure.

In summary, we have developed a modified TSQ mass spectrometer that enabled online observation of the CO$_2$RR and detection of the reaction intermediates and products. The results demonstrated that the coordinated $H_2O$ on Cu(I)-based catalysts played a crucial role in the efficiencies of both $CO_2$ adsorption onto the Cu(I) catalyst and $CO_2$ reduction to CO, and that an improvement of two orders of magnitude was achieved in the presence of a coordinated $H_2O$ than without $H_2O$. Further experiments indicated that the existing form of $H_2O$ and the number of coordinated $H_2O$ also had a significant effect on the reduction of $CO_2$ to CO. More importantly, isotope-labeling investigations revealed that the origin of the O atom in the generated CO originated from $H_2O$, instead of $CO_2$. Based on the experimental observations and computational calculations, the specific pathway for the reduction of $CO_2$ to CO was proposed. This work not only offers a new strategy to disclose the reaction process of CO$_2$RR, but also provides useful insight into the roles of $H_2O$, suggesting the efficiency of CO$_2$RR could be improved by constructing new types of catalysts with coordinated $H_2O$.

## Methods

**Preparation of different Cu-based solutions.** Identical procedures were used to prepare the different Cu-based solutions. Specifically, 0.1 g of Cu-based particles such as CuCl were first dissolved into 1.0 mL of double-deionized water. After sonication for 30 min using a KQ3200DB ultrasonic cleaner (Kunshan Ultrasonic Instrument Co., Ltd., Kunshan, China), 1 μL of the CuCl aqueous solution was added into 999 μL of acetonitrile. Finally, the resulting solution was mixed with a QL-901 Vortex oscillator (Haimen Qilin Beier Instrument Manufacturing Co., Ltd., Haimen, China) for 1 min.

**Preparation of glass capillary with a tip orifice of 1 μm.** A glass capillary with a tip orifice of 1 μm was pulled from borosilicate glass capillary with filament (Sutter Instrument, USA, 1.5 mm o.d., 0.86 mm i.d., 10 cm length) using a micropipette puller (Model P-97, Sutter Instrument Co., Novato. CA, USA). The tip orifice was measured with a metallographic microscope equipped with a DCA 10.0 digital camera (1 million resolution) and had a tip orifice precision of ± 1 μm.

**Online reaction and MS analysis.** All experiments on nanoelectrospray ionization mass spectrometry (nanoESI-MS) were carried out with either a TSQ Quantum Access Max mass spectrometer or an Orbitrap Elite Hybrid Ion Trap-Orbitrap Mass Spectrometer (Thermo Fisher Scientific, San Jose, CA, USA). For nanoESI, 20 μL of a 100 μg mL$^{-1}$ CuCl solution was injected into a glass capillary with 1 μm of tip orifice, and, subsequently, 1.0 kV of DC voltage was applied to the CuCl solution for the generation of various Cu-based ions. The distance between the nanoESI tip and MS inlet capillary was about 10 mm. Mass spectra were recorded in the positive ion mode with a capillary temperature of 270 °C. The identification of Cu-based ions was confirmed by high-resolution mass spectrometry (HRMS) and tandem mass spectrometry (MS/MS) using collision-induced dissociation (CID). Argon gas (99.995% purity) was used as the collision gas. For the reaction between Cu-based ions and $CO_2$, Cu-based ions such as Cu$^+$ and $[Cu(H_2O)]^+$ were first isolated from Q1 and then introduced to Q2 (see Fig. 1). In the reaction unit of Q2, Cu-based ions interacted with $CO_2$ to yield CO under collision energy of 5 V, and the resulting products were subsequently transferred to Q3, followed by detection. It should be pointed out that the collision energy for $CO_2$ reduction was controlled by the operation software of employed commercial mass spectrometer. To explore the effect of acid on the reaction of Cu-based ions and $CO_2$, 2 μL of acetic acid was injected through a six-way valve (Beijing Yijia Technology Co., Ltd., Beijing, China) into the $CO_2$/Ar gas circuit system. To enhance the reaction efficiency between Cu-based ions and $CO_2$, the gas circuit system was heated to different temperatures via a heating tape controlled by an XMT-815 temperature controller (Shanghai Mike Instrument Co., Ltd., Shanghai, China). The actual temperature of heating tape was determined using a 17B Fluke multimeter (Shanghai Fluke Corporation, Shanghai, China).

**Reversed WGSR experiment.** The experiments were carried out with a WGSR apparatus (Tianjin Tongyuan Hengli Technology Co., Ltd, Tianjin, China). The reactor was loaded with 1.0 g of Cu/ZnO/Al$_2$O$_3$ catalyst. The pressures of $H_2^{16}O$ and $H_2^{18}O$ were controlled by a syringe pump with flow rates ranging from 1 to 50 μL min$^{-1}$. After adjusting a flow rate, a balance time of 1.5 h was used to make the equilibrium of $CO_2$ reduction reaction. The air and involved contaminants were replaced with $H_2$ via successive purging. In the reaction process, the reaction partial pressure of $H_2$ and $CO_2$ were kept constant at 2.7 MPa and 0.3 MPa, respectively, and the reactor temperature was maintained at 230 ºC. The gas products were analyzed online with a Fuli GC9790 gas chromatograph equipped with a TDX-01 molecular sieve packing column (2 m x 3 mm). For the $H_2^{18}O$-labelling experiments, the collected gas products were analyzed off-line with an Agilent Technologies 7890B gas chromatograph equipped with a GS-CarbonPLOT capillary column (30 m x 0.32 mm I.D., 1.50 μm film thickness) and an Agilent Technologies 5977 A mass spectrometer. The carrier gas was helium with a flow rate of 15 mL min$^{-1}$. The initial oven temperature was set as 35 °C and maintained for 3 min. Afterward, it was increased to 100 °C with a rate of 40 °C min$^{-1}$ followed by maintaining for 1 min. The injector temperature was 185 °C, and the injection volume was 250 μL with a split ratio of 100:1.

**In situ DRIFTS analysis.** The in situ diffuse reflectance infrared Fourier transform spectroscopy (in situ DRIFTS) results were conducted on an FTIR (Nicolet 6700) equipped with a Pike DRIFT cell (PIKE Technologies) with a KBr window and an MCT/A detector (cooled by liquid nitrogen). The spectra were collected in the range from 900 to 4000 cm$^{-1}$ with 32 scans, and the resolution was 4 cm$^{-1}$. All used cylinder gases were of high purity and dried through a moisture trap (Agilent Technologies) before entering the in situ chamber. To minimize the environment interference, the FTIR chamber was purged with argon (99.999%) in a flow rate of 4 L min$^{-1}$. Prior to the experiment, the sample was treated with hydrogen (99.9999%) in a flow rate of 5 mL min$^{-1}$ for 2 h and then switched to helium (99.9999%) with a flow rate of 5 mL min$^{-1}$ for 2 h at 500 °C to reduce the surface oxide and remove all possible organic contaminants. Subsequently, under the helium flow, the reactor was cooled down to 150 °C or 250 °C for hydrogenation of $CO_2$. When a steady baseline was obtained, hydrogen (3 mL min$^{-1}$) and $CO_2$

(4 mL min$^{-1}$, 5% $CO_2$/95% Ar) were introduced sequentially. To investigate the effect of water on $CO_2$RR, a syringe needle fulfilled with 1 µL of water was inserted into the tube containing hydrogen flow, enabling the introduction of trace water into the reaction system. The employed catalyst, Cu/γ-$Al_2O_3$, was prepared by an incipient-wetness impregnation method using the precursors of Cu($NO_3$)$_2$ (>99.0%, Guangzhou Chemical Reagent Factory, Guangzhou, China) and γ-$Al_2O_3$ powder (99.99%, Energy Chemical Co., Guangzhou, China).

**Electrochemical reduction of $CO_2$ and online mass spectrometer monitoring**. For the electrochemical reduction of $CO_2$ to CO, the reaction products and $CO_2$ were monitored with an online differential electrochemical mass spectrometer (DEMS, Linglu Instrument Co., Ltd, Shanghai, China). The configuration of the electrochemical cell is shown in Supplementary Fig. 18. The cell was fulfilled with 2.5 mL of 0.5 M KCl solution, which was saturated with $CO_2$ gas. During the electrochemical test, a continuous $CO_2$ gas flow was introduced into KCl solution. The working electrode was prepared by sputtering gold nanoparticles on porous membrane. The reference electrode was an Ag/AgCl electrode in saturated KCl solution, and the counter electrode was Pt wire electrode. The linear sweeping voltammetry was conducted on a electrochemical workstation (CHI Instruments, Inc., Austin, TX, USA) with the sweeping rate of 10 mV s$^{-1}$ from −0.8 V to −1.6 V. Simultaneously, the MS signals with the mass/charge ratios of 44·30, and 28 were recorded by the MS. The $^{18}$O-labeled $H_2O$ (purity: 99.0%, $^{18}$O abundance: ≥98.0%) was purchased from Wuhan Isotope Technology Co., Ltd (Wuhan, China).

**Computational details**. All electronic structure calculations were performed with the Gaussian 09 package (revision B.01; Gaussian, Inc., Wallingford CT, 2010)[79]. For geometry optimization, we used the B2PLYP double hybrid density functional method[80] in conjunction with the augmented correlation-consistent polarized triple zeta (aug-cc-pVTZ) basis set with the implicit treatment of scalar-relativistic effects by using the effective core potential (ECP) pseudopotential for the metal atoms[81], and the aug-cc-pVTZ all-electron basis set for all other atoms[82]. Harmonic vibrational frequencies were computed to verify the nature of the stationary points. The minimum structures reported in this work showed only positive eigenvalues of the Hessian matrix, whereas the transition states had one negative eigenvalue. Intrinsic reaction coordinate calculations were also performed to confirm that the transition states were correlated with the designated intermediates[83–86]. The zero-point vibrational energy (ZPVE) and thermal corrections to the enthalpy were calculated for structures optimized at the B2PLYP/cc-pVTZ level. The thermodynamic functions (ΔH) were estimated within the ideal gas, rigid-rotor, and harmonic oscillator approximations at 298 K and 1 atm. For ease of discussion, the symbols "$^{S}$IS" and "$^{S}$FS" are used to describe the initial state (IS), intermediate state (MS), and final state (FS), while "$^{S}$TSm/n" is used for the interconversion transition state between the intermediate states, $^{S}$m and $^{S}$n. The left superscript "S" denotes the spin multiplicity (1 and 3 for singlet and triplet, respectively).

## Data availability

Source data are provided with this paper, which can also be available from the corresponding authors on reasonable request.

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

## Acknowledgements

The authors thank Professor R. Zare at Stanford University, Professor D. Li at China Three Gorges University, and Prof. Y. Lan at Nanjing Normal University for valuable suggestions and Dr. L. Ding at Xi'an Shiyou University for performing the reversed water-gas shift reaction experiments. Z.Z. thanks the National Natural Science Foundation of China for Project 21777128, the Natural Science Basic Research Program of Shaanxi Province of China for Project 2019JC-33, and the Youth Innovation Team of Shaanxi Universities for Project Z19257. Y.Z. thanks the Natural Science Basic Research Program of Shaanxi Province of China for Project 2021GY-247.

## Author contributions

Z.Z., Y.Z., Q.W., and G.Y. devised the initial concept for the work, Z.Z. and Y.Z. designed the experiments, H.Y., R.D., Z.X., F.L., G.Y., and Y.L. carried out the experiments and

analyzed the data. Z.Z., Q.W., Y.Z., G.Y., and Q.M. co-wrote the manuscript. All authors discussed the results and commented on the manuscript at all stages.

## Competing interests

The authors declare no competing interests.
