## [Peer Review File · Nature Communications]

Title: Water Coordinated on Cu(I)-based Catalysts is the Oxygen Source in CO₂ Reduction to COREVIEWER COMMENTS

Reviewer #1 (Remarks to the Author):

This is a very nice manuscript that is suitable for publication in Nature Communications. The significant results are:

- 1) demonstration of role of Cu-bound H₂O on CO₂RR using careful analysis of products via a TS-modified MS spectrometer.
- 2) elucidation of the role of water with isotopic labeling experiments to show that water supplies the O to CO, indicating both O atoms of CO₂ are lost during this reaction. That result is a surprise and their data supports this conclusion. The addition of acid to the reaction also is interesting in its ability to enhance CO₂RR to CO. This result may have strong implications on future catalyst design.
- 3) the above conclusions are further supported with computational studies to propose a mechanistic pathway of this reaction.

My only concern about the proposed mechanism is that a COOH intermediate is formed after the first H⁺/e⁻ addition. If any formic acid or formate is formed from CO₂RR (which easily occurs and is often a strong competing reaction to CO formation in CO₂RR), would it be detected? How do you account for this side reaction in your mechanistic scheme? Also, HER is a competing reaction in aqueous electrochemical CO₂RR. How is the competing reaction accounted for in the mechanistic scheme?

Finally, there are a couple of odd phrases in the text that should be revised (e.g., a pretty amount on page 7, invincible on page 10). Otherwise, very clear and organized presentation of data. Data have error bars - nice.

Reviewer #2 (Remarks to the Author):

In the work titled "Water Coordinated on Cu(I)-based Catalysts is the Oxygen Source in CO₂ Reduction to CO", a modified triple-stage quadrupole mass spectrometer was developed to monitor the CO₂ reduction to CO in the gas phase online, revealing the vital role of H₂O in promoting metal catalysts for "CO₂ reduction". The activity measurements suggested that the coordinated H₂O promoted CO₂ adsorption and subsequent reduction to CO; the isotope-labeling studies gave the evidence that the O atom originated from the coordinated H₂O on catalysts, rather than CO₂ itself. And a detailed reaction mechanism of CO₂ reduction to CO on Cu(I)-based catalysts with coordinated H₂O was proposed based on experimental results and DFT calculations. While the approach taken by the authors shall shed light on a chemical reaction at a fundamental level, the knowledge obtained therein cannot be simply extrapolated to catalytic reactions. More specifically, the present article investigated a very specific system (ionized metal reacting with H₂O and CO₂), but its extension to heterogeneous catalysis needs to be well justified (with a sound instruction that does not mislead the readers). Given other uncertainty listed below, the reviewer would not recommend this article for publication in Nature Communications.

The specific comments are as follows:

1. Throughout the manuscript, the authors called the system of investigation “CO₂ reduction”, however, the overall reaction that the authors investigated remain unclear. When CO₂ is converted into CO, something needs to be oxidized to close the mass-balance. While the authors mentioned H⁺/e⁻ transfer with H⁺ provided from H₂O, the origin of the electron was not documented.
2. The sentence in page 2 “Nevertheless, there have been no reports on the use of MS to study CO₂RR mechanisms in operando.” is not accurate. Actually, MS (DEMS) has been used for CO₂RR mechanisms investigation in e.g., 2018 (Clark et. al., J. Am. Chem. Soc. 2018, 140, 7012).
3. The extension of the finding in the study employing an ionized metal cannot be simply extended to heterogeneous catalysis. In this regard, the description in the manuscript is not accurate. For instance, in Page 4, the authors stated “H₂O molecules on the surface of metal catalysts were effectively active site for facilitating the reduction of CO₂ to CO”. However, their system contains not metal surface but isolated metal atom (with ligated H₂O), which does not allow the authors to conclude anything regarding the “surface” of corresponding metal. Maybe their investigation using metal surface (Figure S16) can support such a claim, however, more detailed investigation in this direction is required for the authors to support the claim.
4. The sentence in page 3 “Cu(I) is the most active species in CO₂RR.” also lacks scientific rigor. It is commonly accepted that the Cu-based electrocatalysts was converted to metallic Cu under reaction condition (Birdja Y. et al., Nat. Energy 2019, 4, 732; Nitopi S. et. al., Chem. Rev. 2019, 119, 7610). Although the surface oxide layer was shown to remain on the surface in some systems, the role of oxidative Cu layer on activity and selectivity remained elusive.
5. When the authors investigated the effect of H₂O on the reduction of CO₂ to CO, the [Cu(H₂O)]⁺, [Ag(H₂O)]⁺, and [Pd(H₂O)]⁺ all could catalyze CO₂ reduction to CO, indicating the coordinated H₂O molecules was the active site. Then what are the roles of transition metals in CO₂ reduction?
6. In Figure 2, is the [Cu(H₂O)]⁺ species stable under increasing pressure and temperature? If not, it is unfair to compare the ration between [Cu(CO₂)]⁺ /Cu⁺ and [Cu(CO₂)]⁺ / [Cu(H₂O)]⁺. The comparison of relative intensity of [Cu(CO₂)]⁺ under different conditions may be more reasonable.
7. The reason of the inability to isolate [Cu(H₂18O)]⁺ by dissolving CuCl into the mixture of acetonitrile and H₂18O was unclear. More detailed explanation on this is required.
8. In Figures 4a and 4b, why did the peak intensity of [63Cu(CO)]⁺ varied significantly under identical conditions?

Reviewer #3 (Remarks to the Author):

In the presented study by Zheng et. al, the authors have used an innovative approach to study the reaction mechanism for CO₂ reduction to form CO. The study is interesting and provides new aspects of the CO₂ reduction reaction, specifically about the source of oxygen when [Cu(H₂O)]⁺ is used as catalyst.

However, the claims presented in this study are too substantial compared to the presented evidence

and, therefore, additional studies are required to provide evidence for the claims. In-situ DRIFTS as well as catalytic test at steady state conditions with dosing of water should be considered.

Also, one should consider that the claims should be revised since they are simply not correct in some sense. For example, in thermal catalytic reduction of CO₂, i.e. CO₂ + H₂ <-> CO + H₂O, the stoichiometric source of oxygen is CO₂. Technically there is no source of oxygen other than CO₂ present in the reaction (except some OH groups on the surface of the support which is completely excluded in this study). Therefore, first the reactants should be written as H₂O+CO₂+H₂ to avoid confusion, and also, the same reaction should be tested at realistic conditions with water dosage and be compared to the one with no water dosage.

Comments:

1- It is stated in the Introduction that: "(i) CO₂ and its resulting products (e.g., CO, CH₄, and CH₃OH) are neutral molecules and are rarely observed in mass spectra". This is simply not correct since all mentioned molecules can be detected in MS! The statement needs to be revised.

2- It is stated that "In contrast to the above routes, herein we discovered for the first time that the source of the O atom in CO originated from H₂O rather than CO₂". As explained above, the stoichiometric source of oxygen is CO₂ (at least in the thermal catalytic CO₂ conversion). Therefore, this statement cannot be entirely true.

3- The method used in the study is referred to as "in-situ reaction mechanism study" but the reaction occurs in Q2 and the outlet stream in Q3 is studied which is not exactly an in-situ study.

4- It is stated that "Considering this study investigated different Cu-based complexes in the reduction of CO₂, nanoESI was utilized to generate Cu(I) species rather than inductively coupled plasma (ICP), 59,60 which only produces Cu⁺." But the oxidation state of Cu(I) is the same as Cu⁺. The statement needs to be revised.

5- It is stated that "we also found that in contrast to Cu⁺, [Cu(H₂O)]⁺ was likely to form Cu-based complexes with other molecules such as methanol (Fig. S7), ethanol (Fig. S8), acetonitrile (Fig. S9), benzene (Fig. S10), toluene (Fig. S11), and dichloromethane (Fig. S12). Such a fact suggests that the coordinated H₂O in the structure of [Cu(H₂O)]⁺ is a uniquely active site for adsorbing different molecules." An important detail missing in the manuscript is to find/study/ or at least speculate the reason for such high adsorption capacity for coordinated H₂O.

6- There is a maximum CO₂ adsorption capacity and maximum CO production rate when [Cu(H₂O)]⁺ is used as catalyst through a range of temperatures and pressures. Why such maximums are achieved? thermodynamically, higher temperatures should increase the CO production. The reason behind observing these maximum values should be studied.

7- In Fig 3, caption needs to be corrected (CO₂ instead of CO should be used).

8- In realistic conditions for heterogeneous thermal catalytic conversion of CO₂ to CO, the ability of H₂O to coordinate with Cu should be tested since it may compete with CO₂ for adsorption sites. It will be important to see the source of oxygen in steady state conditions. Also, since water is a product, addition of water may affect the thermodynamics of the reaction. The steady state conditions (with dosage of water) should be studied and compared with no water dosage to see how it will affect the final yield of CO.

9- It is stated that "...allowed us to draw a definite conclusion that in the CO₂RR process, the two O atoms in CO₂ were eliminated completely, and the O atom in CO originated from the coordinated H₂O." What does eliminated mean here? The statement needs revision to clarify the reaction outcome.

11- the general claim is more substantial than the presented results. The situation is far from actual heterogeneous catalysis and the claims cannot be extended to include the reaction mechanism over supported-Cu based catalysts. The reaction is not studied under realistic conditions. One at least should pair the given results with in-situ DRIFTS as well as catalytic conversion of CO₂ to CO on heterogeneous catalysts to see whether the same results can be achieved.

12- The "reaction temperature" was solely the temperature of the heating tape and not the reaction. This should be clarified in Fig. 2.

For Reviewer #1 (Remarks to the Author)

Comments:

This is a very nice manuscript that is suitable for publication in Nature Communications. The significant results are:

- 1) demonstration of role of Cu-bound H₂O on CO₂RR using careful analysis of products via a TS-modified MS spectrometer.
- 2) elucidation of the role of water with isotopic labeling experiments to show that water supplies the O to CO, indicating both O atoms of CO₂ are lost during this reaction. That result is a surprise and their data supports this conclusion. The addition of acid to the reaction also is interesting in its ability to enhance CO₂RR to CO. This result may have strong implications on future catalyst design.
- 3) the above conclusions are further supported with computational studies to propose a mechanistic pathway of this reaction.

RESPONSE: The authors thanks a lot for the reviewer's valuable comments and suggestions to our work.

My only concern about the proposed mechanism is that a COOH intermediate is formed after the first H⁺/e⁻ addition. If any formic acid or formate is formed from CO₂RR (which easily occurs and is often a strong competing reaction to CO formation in CO₂RR), would it be detected? How do you account for this side reaction in your mechanistic scheme? Also, HER is a competing reaction in aqueous electrochemical CO₂RR. How is the competing reaction accounted for in the mechanistic scheme?

RESPONSE: We understand the reviewer's concern. Formic acid could be captured in our study as the form of [Cu(HCOOH)]⁺ (Fig. S29). In our mechanistic scheme, H₂O molecules or involved acid (e.g., HOAc as Fig. 4b) can release H⁺, which would combine with the intermediate of [Cu(HCOO)] (Fig. 5a) by formation of [Cu(HCOOH)]⁺ as a side reaction.

Fig. S33 | Mass spectra of CO₂ reduction to CO under different Cu-based catalytic systems. (a) [⁶³Cu(H₂O)]⁺ and (b) [⁶⁵Cu(H₂O)]⁺ (reaction temperature: 280 °C; reaction pressure: 1.5 mTorr).

We agree the reviewer's comment that HER is a competing reaction in aqueous electrochemical CO₂RR. In the mechanistic scheme (Scheme 1), H⁺/e⁻ was transferred from the reaction system to the OH groups of ¹MS4 and ²MS5 by the formation of ²MS5 and ¹MS6. In the processes, two molecules of H₂O were lost. For the HER, the reaction involves 2H⁺ + 2e⁻ → H₂, which would compete with the attack of H⁺/e⁻ to ¹MS4 and ²MS5.

If H^+ and e^- were used for HER by formation of H_2 , it would not favor the formation of 2MS5 and 1MS6 . By calculating their corresponding free energy, the values were $-4.76 \text{ kal mol}^{-1}$ and $-60.89 \text{ kal mol}^{-1}$ when 1MS4 and 2MS5 were transferred to 2MS5 and 1MS6 , respectively. For a standard hydrogen electrode, the free energy for the transfer of H^+ to H_2 is $-19.09 \text{ kal mol}^{-1}$ in basic solution, which is the favorable system to aqueous electrochemical CO_2RR while preventing from HER. In contrast to the transfer of 1MS4 to 2MS5 , it was prone to the HER, whereas an opposite trend occurred when 2MS5 was transferred to 1MS6 .

Based on the above, the corresponding description has been made as “...The dehydration process would compete with hydrogen evolution reaction (HER). By calculation of their corresponding free energy, the values were $-4.76 \text{ kal mol}^{-1}$ and $-60.89 \text{ kal mol}^{-1}$ when 1MS4 and 2MS5 were transferred to 2MS5 and 1MS6 , respectively. For a standard hydrogen electrode, the free energy is $-19.09 \text{ kal mol}^{-1}$ as H^+ is transferred to H_2 in basic solution, which system is favorable to aqueous electrochemical CO_2RR while preventing from HER. Apparently, in contrast to the transfer of 1MS4 to 2MS5 , it was prone to the HER, whereas an opposite trend occurred when 2MS5 was transferred to 1MS6 . Along with the processes, 2MS5 would also tend to combine with H^+ by formation of $[Cu(HCOOH)]^+$ as a side reaction, which was captured in our current study (Figs. S33-S34)...” in the 2nd paragraph of Page 10.

Finally, there are a couple of odd phrases in the text that should be revised (e.g., a pretty amount on page 7, invincible on page 10). Otherwise, very clear and organized presentation of data. Data have error bars - nice.

RESPONSE: We thank a lot the reviewer for the advices. These odd phrases have been revised as “a considerable amount” in the 2nd paragraph, Page 7 and “convincing” in the last paragraph, Page 10.

Reviewer #2 (Remarks to the Author)

Comments:

In the work titled “Water Coordinated on Cu(I)-based Catalysts is the Oxygen Source in CO_2 Reduction to CO”, a modified triple-stage quadrupole mass spectrometer was developed to monitor the CO_2 reduction to CO in the gas phase online, revealing the vital role of H_2O in promoting metal catalysts for “ CO_2 reduction”. The activity measurements suggested that the coordinated H_2O promoted CO_2 adsorption and subsequent reduction to CO; the isotope-labeling studies gave the evidence that the O atom originated from the coordinated H_2O on catalysts, rather than CO_2 itself. And a detailed reaction mechanism of CO_2 reduction to CO on Cu(I)-based catalysts with coordinated H_2O was proposed based on experimental results and DFT calculations. While the approach taken by the authors shall shed light on a chemical reaction at a fundamental level, the knowledge obtained therein cannot be simply extrapolated to catalytic reactions. More specifically, the present article investigated a very specific system (ionized metal reacting with H_2O and CO_2), but its extension to heterogeneous catalysis needs to be well justified (with a sound

instruction that does not mislead the readers). Given other uncertainty listed below, the reviewer would not recommend this article for publication in Nature Communications.

RESPONSE: Thanks a lot for this reviewer's valuable comments and suggestions. According to the insightful comments, the replies to them are as below.

The specific comments are as follows:

1. Throughout the manuscript, the authors called the system of investigation "CO₂ reduction", however, the overall reaction that the authors investigated remain unclear. When CO₂ is converted into CO, something needs to be oxidized to close the mass-balance. While the authors mentioned H⁺/e⁻ transfer with H⁺ provided from H₂O, the origin of the electron was not documented.

RESPONSE: In the current investigation, the experiments were carried out with a modified TSQ mass spectrometer. To perform the reaction between [Cu(H₂O)]⁺ and CO₂, we first isolated [Cu(H₂O)]⁺ ions from Q1 in **Scheme 1** and subsequently transferred them to Q2 (reaction region). In the Q2 of mass spectrometer, [Cu(H₂O)]⁺ and CO₂ would react upon the electric field, in which 5 V of voltage was applied to facilitate the reaction. The related description has been made in the 4th paragraph, Page 11 as "...In the reaction unit of Q2, Cu-based ions interacted with CO₂ to yield CO under collision energy of 5 V..." Upon the application of 5 V of voltage, the requested electrons for the reaction would be supplied. To make this clear, this description has been made again in the 3rd paragraph, Page 3 as "...Cu-based ions interacted with CO₂ upon the applied voltage of 5 V..."

2. The sentence in page 2 "Nevertheless, there have been no reports on the use of MS to study CO₂RR mechanisms in operando." is not accurate. Actually, MS (DEMS) has been used for CO₂RR mechanisms investigation in e.g., 2018 (Clark et. al., *J. Am. Chem. Soc.* 2018, 140, 7012).

RESPONSE: According to the reviewer's valuable suggestion, we have revised the corresponding description to "...Nevertheless, there have been few reports on the use of MS to study CO₂RR mechanisms in operando³²..." on lines 14-15, Page 2, and have cited the suggested reference.

3. The extension of the finding in the study employing an ionized metal cannot be simply extended to heterogeneous catalysis. In this regard, the description in the manuscript is not accurate. For instance, in Page 4, the authors stated "H₂O molecules on the surface of metal catalysts were effectively active site for facilitating the reduction of CO₂ to CO". However, their system contains not metal surface but isolated metal atom (with ligated H₂O), which does not allow the authors to conclude anything regarding the "surface" of corresponding metal. Maybe their investigation using metal surface (Figure S16) can support such a claim, however, more detailed investigation in this direction is required for the authors to support the claim.

RESPONSE: Thank you so much for the insightful comments. To make the related description more accurate, we have changed the corresponding statement as "...H₂O molecules on the metal catalysts were active sites for facilitating the effective reduction of CO₂ to CO..." on lines 4-5, Page 4. In addition, more reduction reaction of CO₂ to CO have been performed using other metal catalysts such as Ag and Pd. The corresponding results

are shown in **Figs. S19** and **S20**. It is apparent that H_2O indeed played an important role in CO_2 reduction to CO , which provided more confident evidences to support the claim.

Fig. S19 | Simultaneous measurement of the MS current of different ions using an *in situ* differential electrochemical mass spectrometer: (a) m/z 44, (b) m/z 28, and (c) m/z 30 in the system of H_2^{16}O , and (d) m/z 44, (e) m/z 28, and (f) m/z 30 in the system of H_2^{18}O for Ag electrode in 0.5 M of KHCO_3 aqueous solution.

Fig. S20 | Simultaneous measurement of the MS current of different ions using an *in situ* differential electrochemical mass spectrometer: (a) m/z 44, (b) m/z 28, and (c) m/z 30 in the system of H_2^{16}O , and (d) m/z 44, (e) m/z 28, and (f) m/z 30 in the system of H_2^{18}O for Pd electrode in 0.5 M of KHCO_3 aqueous solution.

By the way, to further confirm the current conclusion in actual heterogeneous catalysis, we employed a water-gas shift reaction (WGSR) apparatus equipped with gas chromatography (Fig. S6a) to *on-line* investigate the effect of H₂O content on the reduction of CO₂ to CO, and commercial available Cu/ZnO/Al₂O₃ particles were used as catalyst. To well control the H₂O content, a certain amount of H₂O was continuously introduced to the reaction system by adjusting the flow rate of syringe pump. As shown in Fig. S6b, the regeneration amount of CO demonstrated a first increasing trend followed by a declining one, in which it gave the optimal performance when the H₂O partial pressure in reaction system was 50 kPa. Such a pattern was in good agreement with the previous theoretical calculations by Sun *et al.* (*Sci. China Chem.* **2019**, 62, 1686-1697). Namely, H₂O could kinetically accelerate the hydrogenation on CO₂ to COOH, promoting the reverse WGSR to produce CO, whereas the too high initial partial pressure of H₂O would thermodynamically inhibit the CO₂ conversion. Although the improvement using the current reaction system was not so impressive, it indicated that the presence of H₂O could indeed promote the conversion of CO₂ to CO in heterogeneous catalysis. The corresponding description has been made as “...The effect of H₂O could also be mirrored by a reverse water-gas shift reaction (WGSR) using Cu/ZnO/Al₂O₃ as catalyst (Fig. S6). It hinted that dosing a suitable amount of H₂O could promote heterogeneous thermal catalytic conversion of CO₂ to CO in realistic conditions...” in Lines 5-8, Page 4.

Fig. S6 | Effect of H₂O pressure on the reduction of CO₂ to CO using a water-gas shift reaction (WGSR) apparatus equipped with gas chromatography: (a) WGSR apparatus; (b) Effect of H₂O pressure on CO generation (reaction temperature: 230 °C; $p(\text{CO}_2) = 0.3$ MPa, $p(\text{H}_2) = 2.7$ MPa, and $p(\text{H}_2\text{O})$ was controlled by the flow rate of syringe pump; $n = 3$).

- The sentence in page 3 “Cu(I) is the most active species in CO₂RR.” also lacks scientific rigor. It is commonly accepted that the Cu-based electrocatalysts was converted to metallic Cu under reaction condition (Birdja Y. *et al.*, *Nat. Energy* 2019, 4, 732; Nitopi S. *et al.*, *Chem. Rev.* 2019, 119, 7610). Although the surface oxide layer was shown to remain on the surface in some systems, the role of oxidative Cu layer on activity and selectivity remained elusive.

RESPONSE: We understand the reviewer’s concern. Based on the reviewer’s suggested references, metallic Cu indeed played crucial roles as electrocatalysts for CO₂RR. By the way, many other references, including *J. Electrochem. Soc.* **158**, E45 (2011); *J. CO₂ Util.* **15**, 96-106 (2016); *J. Am. Chem. Soc.* **142**, 6400-6408 (2020); *J. Am. Chem. Soc.* **143**, 2984-2993 (2021); *J. Am. Chem. Soc.* **142**, 2857-2867 (2020); *Energ. Environ. Sci.* **14**,

1121-1139 (2021); *Chem* **5**, 1818-1833 (2019), also concluded that Cu(I)-based catalysts were critical in CO₂RR. Therefore, to make the statement more sound, this sentence in the 4th paragraph, Page 3 has been revised as “...Cu(I) is one of the most active species in CO₂RR...”

5. When the authors investigated the effect of H₂O on the reduction of CO₂ to CO, the [Cu(H₂O)]⁺, [Ag(H₂O)]⁺, and [Pd(H₂O)]⁺ all could catalyze CO₂ reduction to CO, indicating the coordinated H₂O molecules was the active site. Then what are the roles of transition metals in CO₂ reduction?

RESPONSE: From the current study, the most important role of the transition metals in CO₂ reduction is to provide active sites for the coordination of H₂O. As shown in **Fig. 4**, the coordinated H₂O on the transition metals plays crucial roles in the reduction of CO₂ to CO, rather than free H₂O. The corresponding description has been made as “...The above results also indicated that the coordinated H₂O on the Cu⁺, rather than free H₂O, played crucial roles for the reduction of CO₂ to CO, and the transition metals such as Cu⁺ provided efficient active sites for the generation of [Cu(H₂O)]⁺ or other H₂O-based metal complex ions...” on the 2nd paragraph, Page 8. By the way, such a reduction process may be accompanied by a change in the copper oxidation state, similar to the selective oxidation of methane to methanol (Sushkevich *et al. Science* **2017**, 356, 523-527). However, the variation in the copper oxidation state was not observed in the current condition, which needed to be studied in the future study.

6. In Figure 2, is the [Cu(H₂O)]⁺ species stable under increasing pressure and temperature? If not, it is unfair to compare the ration between [Cu(CO₂)]⁺/Cu⁺ and [Cu(CO₂)]⁺/[Cu(H₂O)]⁺. The comparison of relative intensity of [Cu(CO₂)]⁺ under different conditions may be more reasonable.

RESPONSE: According to the reviewer's valuable suggestions, we investigated the stability of [⁶³Cu(H₂O)]⁺ species under increasing pressure and temperature. As shown in **Fig. S14a**, the peak intensity of [⁶³Cu(H₂O)]⁺ in mass spectrometric analysis remained constant within the reaction pressure range of 0.1 – 1.5 mTorr, while further increasing the reaction pressure from 2.0 to 4.0 mTorr resulted in a steady decreasing trend. An almost same pattern was observed under increasing gas circuit temperatures (**Fig. S14b**). Namely,

Fig. S13 | Variation in the peak intensity of [⁶³Cu(H₂O)]⁺ with reaction pressures and gas circuit temperatures: **(a)** Effect of the reaction pressures on the peak intensity of [⁶³Cu(H₂O)]⁺ (reaction pressure: 1.5 mTorr; n = 5). **(b)** Effect of the gas circuit temperatures on the peak intensity of [⁶³Cu(H₂O)]⁺ (reaction temperature: 280 °C; n = 5).

at a lower temperature (room temperature to 150 °C), the peak intensity of $[^{63}\text{Cu}(\text{H}_2\text{O})]^+$ was relatively stable, whereas it demonstrated a gradual declining pattern within the temperature range of 150 to 350 °C. These results indicated that under increasing pressures and temperatures, the peak intensity of $[^{63}\text{Cu}(\text{H}_2\text{O})]^+$ was not stable in the studied whole range. This phenomenon could be attributed to the fact that by varying the reaction pressure and gas circuit temperature, a portion of $[\text{Cu}(\text{H}_2\text{O})]^+$ would transfer to Cu^+ , $[\text{Cu}(\text{CO}_2)]^+$, and $[\text{Cu}(\text{CO})]^+$ under collision/catalysis in the Q2 (**Scheme 1**) of employed TSQ mass spectrometer. Based on this case, it may be unfair to compare the ration between $[\text{Cu}(\text{CO}_2)]^+/\text{Cu}^+$ and $[\text{Cu}(\text{CO}_2)]^+ / [\text{Cu}(\text{H}_2\text{O})]^+$, as the reviewer's insightful comment. By carefully observing **Fig. 1c, d**, only Cu^+ , $[\text{Cu}(\text{H}_2\text{O})]^+$, $[\text{Cu}(\text{CO})]^+$, and $[\text{Cu}(\text{CO}_2)]^+$ appeared in the mass spectra, and their amounts changed with increasing pressure and temperature. So in the current condition, it might be unreasonable to compare the effects of pressure and temperature on the generation of $[\text{Cu}(\text{CO})]^+$ and $[\text{Cu}(\text{CO}_2)]^+$ using their relative intensity such as $[\text{Cu}(\text{CO}_2)]^+/\text{Cu}^+$ or $[\text{Cu}(\text{CO})]^+ / [\text{Cu}(\text{H}_2\text{O})]^+$.

To address this obstacle, the absolute peak intensity of $[\text{Cu}(\text{CO}_2)]^+$ and $[\text{Cu}(\text{CO})]^+$ in mass spectrometric analysis were employed to evaluate the CO_2 adsorption and CO generation without internal standards or reference, following the previous reports (*Analyst* 2012, **137**, 2556-2558; *Anal. Chem.* 2012, **84**, 931-938). In addition, the current investigation was performed in comparable experimental conditions except for varying pressures and temperatures and, therefore, such a comparison may be more reasonable

Fig. 2 | Variation of CO_2 adsorption and CO generation with reaction pressure and temperature of heating tape around the gas circuit. a,b Effect of reaction pressure on (a) the adsorption of CO_2 onto $^{63}\text{Cu}^+$ and (b) CO generation by CO_2 reduction in the presence of either $^{63}\text{Cu}^+$ or $^{63}\text{Cu}(\text{H}_2\text{O})^+$ catalysts (gas circuit temperature: 280 °C). **c,d** Effect of the gas circuit temperature on (c) the adsorption of CO_2 onto Cu^+ and (d) CO generation by CO_2 reduction in the presence of either $^{63}\text{Cu}^+$ or $^{63}\text{Cu}(\text{H}_2\text{O})^+$ catalysts (reaction pressure: 1.5 mTorr).

than the relative intensity between $[\text{Cu}(\text{CO}_2)]^+$ and $[\text{Cu}(\text{H}_2\text{O})]^+$ /others in the current study. The updated Fig. 2 is shown as below, and the corresponding description has also been revised in Pages 4-6. Despite such a change, the conclusions in the current investigation were not changed.

7. The reason of the inability to isolate $[\text{Cu}(\text{H}_2^{18}\text{O})]^+$ by dissolving CuCl into the mixture of acetonitrile and H_2^{18}O was unclear. More detailed explanation on this is required.

RESPONSE: According to the suggestion, more detailed explanation has been given in the 4th paragraph of Page 7 as “... *This case could be attributable to the fact that there are plenty of H_2^{16}O in air, and the H_2^{18}O in generated $[\text{Cu}(\text{H}_2^{18}\text{O})]^+$ would be exchanged by H_2^{16}O in the plume of nanoESI or the transfer process to the Q1 of mass spectrometer (Scheme 1). As a result, insufficient $[\text{Cu}(\text{H}_2^{18}\text{O})]^+$ ions could be generated and were not favorable for MS detection...*”

8. In Figures 4a and 4b, why did the peak intensity of $[\text{Cu}(\text{CO})]^+$ varied significantly under identical conditions?

RESPONSE: Thanks a lot for the reviewer’s careful comparison. The variation in the peak intensity of Figures 4a and 4b could be attributed that both series of experiments were performed in different periods, which was greatly affected by the nanoESI efficiency and the ion transfer efficiency from ambient condition to vacuum system. To make such a comparison under identical conditions, we carried out both series of experiments again under comparable conditions, and the results were shown in Figures 4a and 4b. It is apparent that the peak intensity of $[\text{Cu}(\text{CO})]^+$ under identical conditions are comparable.

Fig. 4 | Influence of the number of coordinated H₂O, free H₂O, and extraneous acid on the generation of CO. a, Effects of the number of coordinated H₂O and free H₂O on CO₂ reduction to CO under different Cu-based catalytic systems ($[\text{Cu}]^+$, $[\text{Cu}(\text{H}_2\text{O})]^+$, and $[\text{Cu}(\text{H}_2\text{O})_2]^+$). b, Effect of extraneous acid on the reduction of CO₂ to CO under different Cu-based catalytic systems. Note: The free H₂O and HOAc (acetic acid) were injected into the gas circuit system by an injector; reaction temperature: 280 °C; reaction pressure: 1.5 mTorr; n = 5).

Reviewer #3 (Remarks to the Author)

Comments:

In the presented study by Zheng et. al, the authors have used an innovative approach to study the reaction mechanism for CO₂ reduction to form CO. The study is interesting and provides new aspects of the CO₂ reduction reaction, specifically about the source of oxygen when [Cu(H₂O)]⁺ is used as catalyst.

RESPONSE: Thanks a lot for the reviewer's valuable comments.

However, the claims presented in this study are too substantial compared to the presented evidence and, therefore, additional studies are required to provide evidence for the claims. *In-situ* DRIFTS as well as catalytic test at steady state conditions with dosing of water should be considered.

RESPONSE: We really appreciate the reviewer's insightful comments and suggestions. The corresponding reply to the suggestion is given in the responses to Comment 10 as below.

Also, one should consider that the claims should be revised since they are simply not correct in some sense. For example, in thermal catalytic reduction of CO₂, i.e. CO₂ + H₂ → CO + H₂O, the stoichiometric source of oxygen is CO₂. Technically there is no source of oxygen other than CO₂ present in the reaction (except some OH groups on the surface of the support which is completely excluded in this study). Therefore, first the reactants should be written as H₂O+CO₂+H₂ to avoid confusion, and also, the same reaction should be tested at realistic conditions with water dosage and be compared to the one with no water dosage.

RESPONSE: Thank you so much for the reviewer's comments. The corresponding reply to the suggestion is given in the responses to Comment 8 as below.

Comments:

1. It is stated in the Introduction that: "(i) CO₂ and its resulting products (e.g., CO, CH₄, and CH₃OH) are neutral molecules and are rarely observed in mass spectra". This is simply not correct since all mentioned molecules can be detected in MS! The statement needs to be revised.

RESPONSE: According to the insightful comment, the statement has been revised as "...CO₂ and its resulting products (e.g., CO, CH₄, and CH₃OH) are neutral molecules and rarely observed in mass spectra without adding charges through ionization..." in the 2nd paragraph of Page 2.

2. It is stated that "In contrast to the above routes, herein we discovered for the first time that the source of the O atom in CO originated from H₂O rather than CO₂". As explained above, the stoichiometric source of oxygen is CO₂ (at least in the thermal catalytic CO₂ conversion). Therefore, this statement cannot be entirely true.

RESPONSE: To make this statement more exact, we have revised it as "...*In contrast to the above routes, herein we discovered for the first time that the source of the O atom in CO originated from the H₂O coordinated on transition metal-based catalysts (e.g., Cu, Ag, and Pd) rather than CO₂...*" in the 2nd paragraph of Page 3.

3. The method used in the study is referred to as "*in-situ* reaction mechanism study" but the reaction occurs in Q2 and the outlet stream in Q3 is studied which is not exactly

an *in-situ* study.

RESPONSE: Strictly speaking, the method used in the current study should be an "*on-line reaction mechanism study*" rather than an "*in-situ reaction mechanism study*" based on the reviewer's valuable comment. Therefore, "*in-situ*" has been revised as "*on-line*" in the current version, which may be more exact.

4. It is stated that "Considering this study investigated different Cu-based complexes in the reduction of CO₂, nanoESI was utilized to generate Cu(I) species rather than inductively coupled plasma (ICP),^{59,60} which only produces Cu⁺." But the oxidation state of Cu(I) is the same as Cu⁺. The statement needs to be revised.

RESPONSE: As the reviewer's comment, the oxidation state of Cu(I) is the same as Cu⁺. Based on the suggestion, the statement has been revised as "*...Considering that the current study investigated different Cu-based complexes in the reduction of CO₂, nanoESI was utilized to generate Cu(I) species rather than inductively coupled plasma (ICP),^{59,60} which only produces Cu(I)...*" in the 4th paragraph of Page 3.

5. It is stated that "we also found that in contrast to Cu⁺, [Cu(H₂O)]⁺ was likely to form Cu-based complexes with other molecules such as methanol (Fig. S7), ethanol (Fig. S8), acetonitrile (Fig. S9), benzene (Fig. S10), toluene (Fig. S11), and dichloromethane (Fig. S12). Such a fact suggests that the coordinated H₂O in the structure of [Cu(H₂O)]⁺ is a uniquely active site for adsorbing different molecules." An important detail missing in the manuscript is to find/study/or at least speculate the reason for such high adsorption capacity for coordinated H₂O.

RESPONSE: In contrast to Cu⁺, [Cu(H₂O)]⁺ is more likely to form Cu-based complexes with other molecules, but the detailed reasons are not clear so far. To the best of our knowledge, this phenomenon could be speculated as "*...To our knowledge, it could be speculated as the following reasons. As H₂O was bound to Cu⁺, the resulting [Cu(H₂O)]⁺ more likely tended to form hydrogen bond⁶²⁻⁶⁶ or OH- π interactions⁶⁷⁻⁶⁹ with those studied molecules than bare Cu⁺. After undergoing further structural rearrangements, Cu(I)-based complexes were favorably generated. Despite this, detailed reasons need to be further studied...*", which has been made up in the last paragraph of Page 4 and the first paragraph of Page 5.

6. There is a maximum CO₂ adsorption capacity and maximum CO production rate when [Cu(H₂O)]⁺ is used as catalyst through a range of temperatures and pressures. Why such maximums are achieved? Thermodynamically, higher temperatures should increase the CO production. The reason behind observing these maximum values should be studied.

RESPONSE: Thanks a lot for the reviewer's comment. The reason behind observing these maximum values could be associated with the stability of [Cu(H₂O)]⁺ under the studied pressure and temperature ranges (**Fig. S14**). The corresponding description has been made as "*...From the above, it was noticeable that there was a maximum CO₂ adsorption capacity and a maximum CO production rate when [Cu(H₂O)]⁺ was used as catalyst throughout the studied temperature and pressure ranges (**Fig. 2**). This phenomenon was presumably due to the instability of [Cu(H₂O)]⁺ with increasing reaction pressures and*

temperatures. As an example of increasing reaction pressures, the content of CO₂ in the Q2 of mass spectrometer (Scheme 1) steadily increased and therefore, high yields of [Cu(CO₂)]⁺ and [Cu(CO)]⁺ should be generated, whereas an opposite trend was observed within the range of 2 – 4 mTorr (Fig. 2a,b). As aforementioned, [Cu(H₂O)]⁺ was more favorable for the generation of [Cu(CO₂)]⁺ and [Cu(CO)]⁺ than Cu⁺. However, the generated amount of [Cu(H₂O)]⁺ demonstrated a decreasing pattern with increasing reaction pressures from 2 to 4 mTorr (Fig. S14a). A lower amount of [Cu(H₂O)]⁺ would result in a lower generation efficiency to [Cu(CO₂)]⁺ and [Cu(CO)]⁺. As a compromise between reaction pressure and the amount of [Cu(H₂O)]⁺, 1.5 mTorr gave the optimal performance. As to the temperature, the same pattern as the reaction pressure was observed (Fig. 2c,d), and 280 °C offered the highest generation efficiencies to both [Cu(CO₂)]⁺ and [Cu(CO)]⁺. This case may be associated with the compromise of [Cu(H₂O)]⁺ stability (Fig. S14b) and the thermodynamic reaction activity between CO₂ and [Cu(H₂O)]⁺ under increasing temperatures...” in the 2nd paragraph of Page 6.

Fig. S14 | Variation in the peak intensity of [63Cu(H₂O)]⁺ with reaction pressures and gas circuit temperatures: (a) Effect of the reaction pressures on the peak intensity of [63Cu(H₂O)]⁺ (reaction pressure: 1.5 mTorr; n = 5); (b) Effect of the gas circuit temperatures on the peak intensity of [63Cu(H₂O)]⁺ (reaction temperature: 280 °C; n = 5).

7. In Fig 3, caption needs to be corrected (CO₂ instead of CO should be used).

RESPONSE: The caption to Fig. 3 has been corrected, in which “CO” has been revised to “CO₂” as “... **Fig. 3 | Isotope-labeling MS measurement results under different systems. a, [63Cu(H₂¹⁶O)]⁺ and C¹⁶O₂. b, [63Cu(H₂¹⁶O)]⁺, H₂¹⁶O, and C¹⁶O₂. c, [63Cu(H₂¹⁶O)]⁺, H₂¹⁸O, and C¹⁶O₂. d, [63Cu(H₂¹⁶O)]⁺ and C¹⁸O₂. e, [63Cu(H₂¹⁶O)]⁺, H₂¹⁶O, and C¹⁸O₂; f, [63Cu(H₂¹⁶O)]⁺, H₂¹⁸O and C¹⁸O₂ (gas circuit temperature: 280 °C; reaction pressure: 1.5 mTorr)...**” in Page 6.

8. In realistic conditions for heterogeneous thermal catalytic conversion of CO₂ to CO, the ability of H₂O to coordinate with Cu should be tested since it may compete with CO₂ for adsorption sites. It will be important to see the source of oxygen in steady state conditions. Also, since water is a product, addition of water may affect the thermodynamics of the reaction. The steady state conditions (with dosage of water) should be studied and compared with no water dosage to see how it will affect the final yield of CO.

RESPONSE: To the best of our knowledge, this comment is another version of the reviewer’s summarized comment above. In thermal catalytic reduction of CO₂, i.e. CO₂ +

$\text{H}_2 \rightarrow \text{CO} + \text{H}_2\text{O}$ [conventional route (i) in **Scheme 1**], the stoichiometric source of oxygen is indeed CO_2 . The reaction process has been widely recognized (*Chem. Rev.* 1995, **95**, 259-272; *Nat. Commun.* 2017, **8**, 27; *ACS Catal.* 2020, **10**, 11318-11345). As the reviewer's comment, some OH groups on the surface of catalyst support would be involved in the reduction process, and the reactants should be written as $\text{H}_2\text{O} + \text{CO}_2 + \text{H}_2$ to avoid confusion. If so, the reduction reaction would become $\text{H}_2\text{O} + \text{CO}_2 + \text{H}_2 \rightarrow \text{CO} + 2\text{H}_2\text{O}$, in which the role of H_2O in the reactants was ambiguous. Also, the detailed reaction mechanism was not documented and confirmed with related experiments although it may be true. Due to this reason, we did not revise it.

Fig. R1 | Mass spectra of CO_2 reduction to CO with 1:3 CO_2/H_2 under different Cu-based catalytic systems and isotope-labeling MS measurement results using H_2^{16}O and H_2^{18}O . Mass spectra of CO_2 reduction to CO with 1:3 CO_2/H_2 under **A**, $^{63}\text{Cu}^+$, **B**, $[^{63}\text{Cu}(\text{H}_2\text{O})]^+$, and **C**, $[^{63}\text{Cu}(\text{H}_2\text{O})_2]^+$ catalytic systems. **D**, Effect of the number of coordinated H_2O on CO_2 reduction to CO under different Cu-based catalytic systems ($^{63}\text{Cu}^+$, $[^{63}\text{Cu}(\text{H}_2\text{O})]^+$, and $[^{63}\text{Cu}(\text{H}_2\text{O})_2]^+$). Isotope-labeling MS measurement results by injecting **E**, H_2^{16}O and **F**, H_2^{18}O into the reaction system of $[^{63}\text{Cu}(\text{H}_2\text{O})]^+$ and CO_2/H_2 (1:3). (reaction temperature: 280 °C; reaction pressure: 1.5 mTorr).

According to the reviewer's suggestion, we performed the reaction of $\text{CO}_2 + \text{H}_2$ using our current experimental conditions, in which 1:3 of CO_2/H_2 (v/v) mixture was used as reactants to react with Cu^+ , $[^{63}\text{Cu}(\text{H}_2\text{O})]^+$, and $[^{63}\text{Cu}(\text{H}_2\text{O})_2]^+$. As displayed in **Fig. R1A**, no obvious CO-related products could be observed in the mass spectrum for the system with bare Cu^+ , whereas the products $[^{63}\text{Cu}(\text{CO})]^+$ (**Fig. R1B**) and $[^{63}\text{Cu}(\text{H}_2\text{O})(\text{CO})]^+$ (**Fig. R1C**) appeared when $[^{63}\text{Cu}(\text{H}_2\text{O})]^+$ and $[^{63}\text{Cu}(\text{H}_2\text{O})_2]^+$ were employed as catalysts, respectively. This indicated that the coordinated H_2O on Cu^+ played a crucial role in determining the reduction reaction of CO_2 and H_2 to CO, in good agreement with the reviewer's envision. Moreover, the generation efficiency of CO varied with the number of the coordinated H_2O on Cu^+ , and $[^{63}\text{Cu}(\text{H}_2\text{O})]^+$ gave the optimal performance (**Fig. R1D**). Such a result illustrated that in the reduction of CO_2 and H_2 to CO, there was an optimal number of coordinated H_2O , in accordance with the previous reports on investigating the effect of H_2O to methanol generation using CO_2 and Cu catalysts (*J. Catal.* 2013, **298**, 10-17; *Fuel Sci. Tech. Int.* 1989, **7**, 899-918).

In realistic conditions for heterogeneous thermal catalytic conversion of CO₂ to CO, the ability of H₂O to coordinate with Cu may compete with CO₂ for adsorption sites. After theoretical calculations, the free energy of Cu⁺ + H₂O → [Cu(H₂O)]⁺ is -40.18 kcal/mol, whereas the value for Cu⁺ + CO₂ → [Cu(CO₂)]⁺ is only -19.68 kcal/mol. Apparently, it is more favorable to form [Cu(H₂O)]⁺ than [Cu(CO₂)]⁺, which may be the reason that a more intensive peak was observed for [Cu(H₂O)]⁺ in mass spectra than for [Cu(CO₂)]⁺ (Fig. R1B). These results indicated that in the catalytic conversion of CO₂ to CO, the coordination ability of H₂O to Cu⁺ was stronger than that of CO₂.

To study the source of oxygen in the generated CO, we introduced 1 μL of H₂¹⁶O and H₂¹⁸O into the reaction system of CO₂, H₂, and [⁶³Cu(H₂¹⁶O)]⁺, respectively, similar to that for Fig. 3b,c,e,f in the main text. As shown in Fig. R1E, the major reduction product of CO₂ and H₂ was [⁶³Cu(C¹⁶O)]⁺ when H₂¹⁶O was introduced. However, both [⁶³Cu(C¹⁶O)]⁺ and [⁶³Cu(C¹⁸O)]⁺ emerged in the mass spectrum as H₂¹⁸O was added. These results demonstrated that in the reduction of CO₂ using 1:3 of CO₂/H₂ as the reactants, the source of oxygen was from involved H₂O, rather than CO₂. Overall, these information supported enough evidences that H₂O could promote the catalytic reaction of CO₂/H₂, but it was out of the scope of current investigation (CO₂ reduction to CO without the involvement of H₂). Therefore, the related content was not incorporated into the manuscript.

Fig. S6 | Effect of H₂O pressure on the reduction of CO₂ to CO using a water-gas shift reaction (WGSR) apparatus equipped with gas chromatography: (a) WGSR apparatus; (b) Effect of H₂O pressure on CO generation (reaction temperature: 230 °C; $p(\text{CO}_2) = 0.3$ MPa, $p(\text{H}_2) = 2.7$ MPa, and $p(\text{H}_2\text{O})$ was controlled by the flow rate of syringe pump; $n = 3$).

To understand the effect of water dosage on the generation of CO from CO₂ in actual heterogeneous catalysis, we used a water-gas shift reaction (WGSR) apparatus equipped with gas chromatography (Fig. S6a) to *on-line* investigate the effect of H₂O content on the reduction of CO₂ to CO, and commercial available Cu/ZnO/Al₂O₃ was used as catalyst. To well control the H₂O content, a certain amount of H₂O was continuously introduced to the reaction system by adjusting the flow rate of syringe pump. As shown in Fig. S6b, the CO regeneration demonstrated a first increasing trend followed by a declining one, in which it gave the optimal value when the H₂O partial pressure in reaction system was 50 kPa, namely 5 μL min⁻¹. Such a pattern was in good agreement with the previous theoretical calculations by Sun *et al.* (*Sci. China Chem.* **2019**, 62, 1686-1697). Namely, H₂O could kinetically accelerate the hydrogenation on CO₂ to COOH, promoting the reverse WGSR to produce CO, whereas a too high partial pressure of H₂O would thermodynamically inhibit

the CO₂ conversion. Although the improvement using the current reaction system was not so impressive, it indicated that the presence of H₂O could indeed promote the conversion of CO₂ to CO. In our opinion, more interesting results could be generated if the background H₂O in the reaction system was thoroughly removed, which is the direction of future endeavor. The corresponding description has been made as “...The effect of H₂O could also be mirrored by a reverse water-gas shift reaction (WGSR) using Cu/ZnO/Al₂O₃ as catalyst (Fig. S6). It hinted that dosing a suitable amount of H₂O could promote heterogeneous thermal catalytic conversion of CO₂ to CO in realistic conditions...” in Lines 5-8, Page 4.

9. It is stated that “...allowed us to draw a definite conclusion that in the CO₂RR process, the two O atoms in CO₂ were eliminated completely, and the O atom in CO originated from the coordinated H₂O.” What does eliminated mean here? The statement needs revision to clarify the reaction outcome.

RESPONSE: Based on the suggestion, this statement has been revised as “...allowed us to draw a definite conclusion that in the CO₂RR process, the two O atoms in CO₂ were eliminated completely by losses of two molecules of H₂O,^{46,48,70} and the O atom in CO originated from the coordinated H₂O...” in the 2nd paragraph of Page 7.

10. The general claim is more substantial than the presented results. The situation is far from actual heterogeneous catalysis and the claims cannot be extended to include the reaction mechanism over supported-Cu based catalysts. The reaction is not studied under realistic conditions. One at least should pair the given results with *in-situ* DRIFTS as well as catalytic conversion of CO₂ to CO on heterogeneous catalysts to see whether the same results can be achieved.

RESPONSE: As the reviewer's comment, the general claim is far from actual heterogeneous catalysis. To address this concern, we employed a water-gas shift reaction (WGSR) apparatus equipped with gas chromatography to *on-line* investigate the effect of H₂O content on the reduction of CO₂ to CO, and commercial available Cu/ZnO/Al₂O₃ was used as catalyst. The corresponding description has been made as the reply to Comment 8. From the results, it could be concluded that the current claim was applicable to actual heterogeneous catalysis although the performance was not so impressive using the current apparatus.

According to the reviewer's suggestion, *in-situ* DRIFTS could give more convincing results to the current claim. Based on this, the experiment was carried out at 150 °C and 250 °C with Cu/ γ -Al₂O₃ as catalyst, respectively, and the corresponding results are displayed in Fig. S34. It is apparent that for both reaction temperatures, the *in situ* DRIFT spectra had an analogous pattern by adding H₂O into the reaction system. Namely, after introducing CO₂ into the system (without H₂O), the absorption peak intensity of bicarbonate on γ -Al₂O₃ (1230 cm⁻¹, 1439 cm⁻¹, and 1670 cm⁻¹) demonstrated a gradual increasing trend and reached to a plateau. Simultaneously, the absorption peak of bidentate formate (1602 cm⁻¹) on γ -Al₂O₃ steadily emerged, but this type of absorption peak on Cu (1649 cm⁻¹) was not obvious. However, after H₂O was introduced to the reaction system, on the one hand, the absorption peak of bidentate formate on γ -Al₂O₃ (1379 cm⁻¹, 1398 cm⁻¹, and 1602 cm⁻¹)

Fig. S34 | *In situ* DRIFT spectra of the resulting products on Cu/ $\gamma\text{-Al}_2\text{O}_3$ with addition of H₂O into the reaction system of CO₂ and H₂ at different reaction temperatures: (a) 150 °C and (b) 250 °C (flow rate of 5% CO₂/95% Ar: 4 mL min⁻¹; flow rate of H₂: 3 mL min⁻¹).

Fig. R2 | *In situ* DRIFT spectra of the resulting products on Pt/ $\gamma\text{-Al}_2\text{O}_3$ with addition of H₂O into the reaction system of CO₂ and H₂ (reaction temperature: 50 °C; flow rate of 5% CO₂/95% Ar: 4 mL min⁻¹; flow rate of H₂: 3 mL min⁻¹).

Note: This figure is from the data of our ongoing work in preparation for publication. The figure is only for the response to the reviewer's comments but not for publication.

1) gradually increased, along with the gradual decay till disappearance of the absorption peak of bicarbonate occurred at 1230 cm⁻¹ and 1439 cm⁻¹. More importantly, the absorption peak of bidentate formate on Cu (1649 cm⁻¹) exhibited a gradual increasing trend. These results indicated that H₂O played a crucial role in the generation of formate on both Cu and $\gamma\text{-Al}_2\text{O}_3$. According to our discussion on **Fig. 5a**, formate was a critical intermediate in the reduction of CO₂ to CO, which indirectly confirmed the critical role of H₂O in CO₂RR. Despite this, no obvious absorption peaks of CO adsorbed on either $\gamma\text{-Al}_2\text{O}_3$ or Cu were observed, due to its fast desorption rate or weak adsorption on both supports under the

current conditions (*Catal. Sci. Technol.* **2013**, 3, 767-778).

To get more compelling results on the effect of H₂O in CO₂RR, Pt/ γ -Al₂O₃ was employed as a catalyst to perform the experiments. As shown in **Fig. R2**, after introducing H₂O into the reaction system, the absorption peak of CO adsorbed on Pt (2000 cm⁻¹ and 2061 cm⁻¹) demonstrated a gradual increasing pattern with extension of reaction period. Such a result indicated that the introduced H₂O was indeed favorable to the generation of CO in CO₂RR.

11. The "reaction temperature" was solely the temperature of the heating tape and not the reaction. This should be clarified in Fig. 2.

RESPONSE: According to the valuable suggestion, we have revised "reaction temperature" to "gas circuit temperature" as **Fig. 2**. The corresponding description has been clarified as "...at different temperatures, which was controlled by varying the temperature of heating tape around the gas circuit (**Scheme 1**)..." in the 3rd paragraph of Page 5.

Fig. 2 | Variation of CO₂ adsorption and CO generation with reaction pressure and temperature of heating tape around the gas circuit. a,b, Effect of reaction pressure on (a) the adsorption of CO₂ onto ⁶³Cu⁺ and (b) CO generation by CO₂ reduction in the presence of either ⁶³Cu⁺ or [⁶³Cu(H₂O)]⁺ catalysts (gas circuit temperature: 280 °C). **c,d**, Effect of the gas circuit temperature on (c) the adsorption of CO₂ onto Cu⁺ and (d) CO generation by CO₂ reduction in the presence of either ⁶³Cu⁺ or [⁶³Cu(H₂O)]⁺ catalysts (reaction pressure: 1.5 mTorr).

REVIEWER COMMENTS

Reviewer #2 (Remarks to the Author):

In the updated manuscript, the authors provided some additional supporting data to claim that the coordinated H₂O promoted CO₂ adsorption and subsequent reduction, and the O atom originated from the coordinated H₂O on catalysts, rather than CO₂ itself. Unfortunately, this reviewer's concerns were not resolved fully. Specifically, the observed results of CO₂ reduction in this triple-stage quadrupole mass spectrometer cannot distinguish either electrochemical CO₂ reduction or reversed water-gas shift reaction (RWGS is also considerable to be CO₂ reduction reaction). The gap between this condition (TSQ mass spectrometer) and realistic heterogeneous catalysis is unclear. The present work investigated a very specific system (ionized metal reacting with H₂O and CO₂), but its extension to heterogeneous catalysis needs to be well justified (with a sound instruction that does not mislead the readers). Given other uncertainty listed below, the reviewer would not recommend this article for publication in Nature Communications as it currently stands.

The specific comments are as follows:

1. The comparability of the reaction condition in between TSQ mass spectrometer and realistic thermal or electrochemical catalysis should be provided.
2. The CO₂ reduction reaction occurred in Q2, where the temperature was 280 °C and the applied voltage was 5 V. How did the authors isolate the contributions of temperature and voltage?
3. Throughout the manuscript, the authors called the system of investigation "CO₂ reduction", the overall reaction that the authors investigated remain unclear. When CO₂ is converted into CO, something needs to be oxidized to close the mass balance. For example, in thermal catalytic reduction, H₂ is usually needed; in electrochemical CO₂ reduction, H₂ was easily formed. However, there was no H₂ introduction or generation mentioned in this work.
4. Although the authors employed a water-gas shift reaction (WGS) apparatus) to on-line investigate the effect of H₂O content on the reduction of CO₂ to CO. The conclusion of "Optimized amount of H₂O could enhance the hydrogenation on CO₂ to produce CO" was well-known. On the contrary, the evidence of one conclusion claimed in this work "O atom originated from the coordinated H₂O on catalysts, rather than CO₂ itself" was still missing. Additionally, the oxygen-exchange on metal-support interfaces during CO₂ hydrogenation also was reported (J. Catal. 2018, 367, 194). So, this part does not bring anything concrete.
5. In Page 3, the revised version, the sentence "Cu(I) is one of the most active species in CO₂RR" still lacks scientific rigor. The metallic Cu was already observed under reaction condition during electrochemical CO₂ reduction, while the surface oxide species usually resulted in the different product distributions (Birdja Y. et al., Nat. Energy 2019, 4, 732; Nitopi S. et. al., Chem. Rev. 2019, 119, 7610).
6. In the "rebuttal letter", the authors replied that the most important role of the transition metals in CO₂ reduction is to provide active sites for the coordination of H₂O. CO₂ reduction reaction occurs on various metals even over the p-block elements. What is the oxidation state of transition metals during the CO₂ reduction, persisting "+1"? Are there any other roles of transition metals in CO₂ reduction?

7. In Figure 4, the number of coordinated H₂O has less effect on CO generation. If the [63Cu(H₂O)]⁺ was the active site for CO₂ reduction to CO, two H in coordinated H₂O were needed and the O in coordinated H₂O was in CO. Then the pathways of O in CO₂ was not documented.

Reviewer #3 (Remarks to the Author):

Zheng et al. have considered the comments by this reviewer and implemented the necessary changes. One comment to address is that, while the thermal heterogeneous catalysis test was done using various water partial pressure, the comparison between the total conversion with and without H₂O dosage is missing. A graph on CO₂ conversion under both conditions (with and without steady state dosage of H₂O) should be added to the manuscript to clearly show the effect of water addition on the overall performance of the catalyst.

For Reviewer #2 (Remarks to the Author)

Comments:

In the updated manuscript, the authors provided some additional supporting data to claim that the coordinated H₂O promoted CO₂ adsorption and subsequent reduction, and the O atom originated from the coordinated H₂O on catalysts, rather than CO₂ itself. Unfortunately, this reviewer's concerns were not resolved fully. Specifically, the observed results of CO₂ reduction in this triple-stage quadrupole mass spectrometer cannot distinguish either electrochemical CO₂ reduction or reversed water-gas shift reaction (RWGS is also considerable to be CO₂ reduction reaction). The gap between this condition (TSQ mass spectrometer) and realistic heterogeneous catalysis is unclear. The present work investigated a very specific system (ionized metal reacting with H₂O and CO₂), but its extension to heterogeneous catalysis needs to be well justified (with a sound instruction that does not mislead the readers). Given other uncertainty listed below, the reviewer would not recommend this article for publication in *Nature Communications* as it currently stands.

RESPONSE: The authors thank very much to the reviewer for the satisfaction about concerns from the initial review and kind reminds from second review. Based on the insightful comments, the replies to the raised concerns are as below.

The specific comments are as follows:

1. The comparability of the reaction condition in between TSQ mass spectrometer and realistic thermal or electrochemical catalysis should be provided.

Table S4 | Comparison of the reaction conditions among TSQ mass spectrometer, realistic thermal (reverse water-gas shift reaction) and electrochemical catalysis.

Reaction System	Reaction Phase	Bulk Phase	Gas Component	Catalyst	Temperature	Pressure
TSQ system	gas phase	Ar gas	5% (v/v) CO ₂ /Ar	Cu, Ag or Pd-related ions	280 °C ☆	1.3 x 10 ⁻⁷ MPa (1 mTorr)
realistic thermal catalysis	gas/solid phase	H ₂ gas	9:1 (p/p) H ₂ /CO ₂	Cu/ZnO/Al ₂ O ₃	230 °C	3.0 MPa
electrochemical catalysis	gas/liquid phase	Ar gas/KCl solution	100% CO ₂	Ag/Au/Pd electrodes	room temperature	0.1 MPa (1 atm)

Note: ☆ means the gas circuit temperature, rather than the reaction cell Q2 temperature, as shown in Scheme 1.

RESPONSE: Thanks a lot for the reviewer's comment. According to the valuable suggestion, the comparison of the reaction conditions among TSQ mass spectrometer, realistic thermal (reverse water-gas shift reaction) and electrochemical catalysis has been provided in **Table S4**, which may be helpful for readers to understand their differences. From this table, it is apparent that the reaction conditions have much difference for CO₂RR. Despite this, the common point among them is that all the reactions between CO₂ and H₂O occur at the metal-based catalyst interfaces, which paves the way for studying the effect of H₂O on the efficiency of CO₂RR. The corresponding content has been added in the Supporting Information.

2. The CO₂ reduction reaction occurred in Q2, where the temperature was 280 °C and the applied voltage was 5 V. How did the authors isolate the contributions of temperature

and voltage?

RESPONSE: As the temperature of reaction cell Q2 in mass spectrometer could not be regulated, the contribution of temperature was isolated by adjusting the gas circuit temperature controlled by a heating tape, as shown in the blue arrow of **Scheme 1**. The corresponding description has been made as “...*A similar process was also conducted to compare the adsorption of CO₂ and generation of CO at different temperatures, which was controlled by varying the temperature of heating tape around the gas circuit (Scheme 1)*...” in the last paragraph of Page 5.

Scheme 1 | Scheme of the apparatus for CO₂ reduction and detection of reaction products (insets in the top left corner are the different routes for generation of CO from CO₂, in which * means catalyst).

The contribution of voltage in Q2 was controlled by the operation software of employed commercial TSQ mass spectrometer, as demonstrated in the red region of **Fig. R1**. The related content has been made up as “...*It should be pointed out that the collision energy for CO₂ reduction was controlled by the operation software of employed commercial mass spectrometer*...” in the 1st paragraph of Page 12.

Fig. R1 | Operation panel of employed TSQ Quantum Access Max Mass Spectrometer.

3. Throughout the manuscript, the authors called the system of investigation “CO₂ reduction”, the overall reaction that the authors investigated remain unclear. When CO₂ is converted into CO, something needs to be oxidized to close the mass balance. For example, in thermal catalytic reduction, H₂ is usually needed; in electrochemical CO₂ reduction, H₂ was easily formed. However, there was no H₂ introduction or generation

mentioned in this work.

RESPONSE: We really appreciate the reviewer's insightful comment. According to our proposed reaction equation (iv) in **Scheme 1**, namely $*\text{H}_2\text{O} + \text{CO}_2 + 2\text{e}^- + 2\text{H}^+ \rightarrow *\text{CO} + 2\text{H}_2\text{O}$ (* means metal ions such as Cu^+), H^+ ions originated from the dissociation of $\text{H}_2\text{O}/\text{H}_2\text{CO}_3$ or added acid, and the necessary electrons involved in the reduction of CO_2 to CO were likely from the generated H_2CO_3 .

As is well-known, the Q2 region of commercial TSQ mass spectrometer (**Scheme 1**) is generally employed for collision-induced dissociation (CID) of gas phase ions, in which the selected ions are accelerated by applying an electrical potential (5 V of AC voltage in this work) to increase the ion kinetic energy and then allowed to collide with neutral molecules (e.g., argon in this work). In the collision, some of the kinetic energy is converted into internal energy which results in bond breakage and the fragmentation of the molecular ion into smaller fragments ([https://en.wikipedia.org/wiki/collision-induced dissociation](https://en.wikipedia.org/wiki/collision-induced_dissociation)). In the current investigation, it involves the reactants of $[\text{Cu}(\text{H}_2\text{O})]^+$, Cu^+ , H_2O , and CO_2 in the Q2 region during the reaction/collision between $[\text{Cu}(\text{H}_2\text{O})]^+$ and CO_2 . Along with them, $[\text{Cu}(\text{H}_2\text{CO}_3)]^+$ or H_2CO_3 would be generated upon the collision between $[\text{Cu}(\text{H}_2\text{O})]^+/\text{H}_2\text{O}$ and CO_2 (**Fig. S23**). Subsequently, on the one hand, H_2CO_3 dissociates through Eqs. (1) and (2), and the resulting H^+ ions would supply the necessary protons in **Fig. 5a** in the main text. On the other hand, many prior studies [*Chem. Rev.* **2002**, 102, 231–282; *Phys. Scr.* **2007**, 76 C56–C62; *J. Chem. Phys.* **2002**, 116, 6560–6566; *Chem. Phys. Lett.* **1984**, 106, 544–549; *Planet. Space Sci.* **1981**, 29, 735–739] have indicated that electrons could be dissociated from anions [e.g., PtBr_6^{2-} , $\text{Pd}(\text{CN})_4^{2-}$, $\text{Ru}^3\text{Co}(\text{CO})^{13-}$, CO_3^- , and CO_2^-] in the CID of mass spectrometry. Based on the above fact, it is speculated that the required electrons in this study could be generated following Eqs. (3)–(5) in the dissociation of CO_3^{2-} .

If the above assumption was correct, O_2 should be produced. After examining the mass spectrum from the reaction of $[\text{Cu}(\text{H}_2\text{O})]^+$ and CO_2 , $[\text{Cu}(\text{O}_2)]^+$ ions, namely m/z 95 for $[\text{Cu}(\text{O}_2)]^+$ and 97 for $[\text{Cu}(\text{O}_2)]^+$, were indeed captured as shown in **Fig. S27a** and **b**. The information suggested that in CO_2RR , the dissociation of H_2CO_3 was likely to be one of the electron sources.

In addition, we also explored the possibility of generating necessary electrons from Cu-based species in the CID, namely Eq. (6) or (7). If this route was feasible, a higher collision energy in CID would favor a more amount of electrons, thereby leading to a decrease in the total amount of Cu(I)-based species (e.g., $[\text{Cu}(\text{H}_2\text{O})]^+$, Cu^+ , $[\text{Cu}(\text{CO})]^+$, and $[\text{Cu}(\text{CO}_2)]^+$). Otherwise, the amount of Cu-based species would keep constant.

To confirm the above possibility, in the reaction between $[\text{Cu}(\text{H}_2\text{O})]^+$ and CO_2 we enhanced the collision energy ranging from 4 to 20 V by maintaining other parameters

constant. With the increase in the collision energy, the dissociation possibility of Cu^+ to Cu^{2+} or $[\text{Cu}(\text{H}_2\text{O})]^+$ to $[\text{Cu}(\text{H}_2\text{O})]^{2+}$ would increase. To evaluate the total amount of Cu-based species, we summed the peak intensity of $[\text{Cu}(\text{H}_2\text{O})]^+$, Cu^+ , $[\text{Cu}(\text{CO})]^+$, and $[\text{Cu}(\text{CO}_2)]^+$ collected from the corresponding mass spectra. As shown in **Fig. S27c** and **d**, the amount of Cu(I)-based species presented a decreasing trend with increasing the collision energy, indicating that in the reaction of $[\text{Cu}(\text{H}_2\text{O})]^+$ and CO_2 , Cu(I)-based species would become others such as Cu(II)-based species via Eqs. (6) and (7), metallic Cu by reacting with the generated electrons from H_2CO_3 or transferring their charges to Ar gas by generation of Ar^+ (m/z 40). However, no direct evidence was gained to confirm the generation of Cu(II)-based species and Ar^+ using the current technique. In our opinion, the reduction of Cu(I)-based species to metallic Cu was highly possible because the required electrons were available in the current system, as well as the documented references (Kim *et al.*, *J. Am. Chem. Soc.* **2003**, 125, 10684-10692; Kim *et al.*, *Phys. Chem. Chem. Phys.* **2015**, 17, 824-830).

From the above discussion, it is apparent that there is at least more than one route to generate necessary electrons for supplying CO_2RR in the current study. Although there was no H_2 introduction or generation involved in the current work, the electrons offered opportunity to reduce CO_2 to CO . The above content has been added in the Supporting Information.

Fig. S27 | Mass spectra of the products by interaction between $[\text{Cu}(\text{H}_2\text{O})]^+$ and CO_2 and variation in the peak intensity of different Cu-based species with increasing collision energy: Mass spectra of the products by interaction (a) between $[\text{Cu}(\text{H}_2\text{O})]^+$ and CO_2 and (b) between $[\text{Cu}(\text{H}_2\text{O})]^+$ and CO_2 ; Variation in the peak intensity of different (c) ^{63}Cu -based and (d) ^{65}Cu -based species with increasing collision energy (gas circuit temperature: 280 °C; reaction pressure: 1.5 mTorr).

4. Although the authors employed a water-gas shift reaction (WGSR) apparatus) to on-line investigate the effect of H_2O content on the reduction of CO_2 to CO . The conclusion of “Optimized amount of H_2O could enhance the hydrogenation on CO_2 to produce CO ”

was well-known. On the contrary, the evidence of one conclusion claimed in this work “O atom originated from the coordinated H₂O on catalysts, rather than CO₂ itself” was still missing. Additionally, the oxygen-exchange on metal-support interfaces during CO₂ hydrogenation also was reported (*J. Catal.* 2018, 367, 194). So, this part does not bring anything concrete.

RESPONSE: The authors agree the reviewer's point. As requested by other reviewer, we carried out the realistic thermal catalysis experiment using water-gas shift reaction (WGS) apparatus. Indeed, the results gave insight into the effect of H₂O content on the reduction of CO₂ to CO but did not bring concrete to the conclusion “O atom originated from the coordinated H₂O on catalysts, rather than CO₂ itself.” In the suggested reference (*J. Catal.* 2018, 367, 194-205), Yan *et al.* pointed out that the interfacial O species in Ru-O-Al interfaces played a critical role in CO₂ activation. Namely, the exchange of O atom occurred between the Ru-O-Al interfaces and the feeding CO₂, consequently by incorporating the interfacial O atom into the final hydrogenation product. However, if the interfacial O atom was from H₂O, it was still difficult to directly confirm the sources of interfacial O species in Ru-O-Al interfaces, namely from adsorbed H₂O or coordinated H₂O. To the best of our knowledge, combining mass spectrometric analysis and catalytic reaction might be an effective avenue to unveil the inside story. The current investigation might be a typical example to demonstrate this point.

Because the heterogeneous catalysis in gas phase could mirror the importance of H₂O content on CO₂RR, as a compromise we have reworded the related description as “...The effect of H₂O could also be mirrored by a reverse water-gas shift reaction (WGS) using Cu/ZnO/Al₂O₃ as catalyst (Fig. S6). It hinted that dosing a suitable amount of H₂O could promote heterogeneous thermal catalytic conversion of CO₂ to CO in realistic conditions...” in the 1st paragraph of Page 4. The content might be helpful to future methodology development in realistic thermal catalysis of CO₂RR and therefore, is not removed from the manuscript.

5. In Page 3, the revised version, the sentence “Cu(I) is one of the most active species in CO₂RR” still lacks scientific rigor. The metallic Cu was already observed under reaction condition during electrochemical CO₂ reduction, while the surface oxide species usually resulted in the different product distributions (Birdja Y. *et al.*, *Nat. Energy* 2019, 4, 732; Nitopi S. *et. al.*, *Chem. Rev.* 2019, 119, 7610).

RESPONSE: Thank the reviewer very much for the comment. In the electrochemical reduction of CO₂, Cu-based catalysts are the most widely investigated materials due to the diversity of the reduction products, which has been extensively discussed in the two classical review articles mentioned by the reviewer.

The subject of Cu-related materials as active species for CO₂RR has been widely investigated in literature, such as *J. Catal.* 1979, 56(3), 407-429; *Appl. Catal.* 1986, 25(1-2), 101-107; *Catal. Lett.* 1994, 30, 99-111; *Appl. Catal. A* 2001, 218(1-2), 235-240; *Surf. Sci.* 2007, 601(14), 3125-3132; *J. Electrochem. Soc.* 2011, 158(2), E45-E49; *Nat. Commun.* 2016, 7(1), 12123; *Nat. Catal.* 2018, 1(2), 103-110; *Nat. Commun.* 2018, 9(1), 3828. Taken together, Cu(I) species played a crucial role in determining the performance of CO₂RR. After studying the suggested references (*Nat. Energy* 2019, 4, 732; *Chem. Rev.* 2019, 119,

7610), they also demonstrated the significance of Cu(I) species in CO₂ reduction. For example, Birdja *et al* (*Nat. Energy* **2019**, *4*, 732-745) stated that “...It is often assumed that during CO₂RR, the Cu_xO electrocatalyst should be completely converted to metallic Cu in CO₂RR. However, a residual oxide layer was shown to remain present on the surface during CO₂RR and the surface oxide and OD metallic layer were proposed as key reaction sites for catalysis¹²⁵. The formation of C3-C4 was also reported, supposedly due to the synergistic effect between Cu₂O and C1 adsorption, which resulted in a higher population of Cu⁺ species...” in the left panel of Page 738. Although the roles of different valences of copper in CO₂RR still remain the subject of debate, many evidences have indicated the roles of Cu(I) species. In combination of the valuable suggestion from the reviewer, we have revised the commented sentence as “Cu(I)-related species are one of the most important catalysts in CO₂RR” in the 4th paragraph of Page 3.

6. In the “rebuttal letter”, the authors replied that the most important role of the transition metals in CO₂ reduction is to provide active sites for the coordination of H₂O. CO₂ reduction reaction occurs on various metals even over the *p*-block elements. What is the oxidation state of transition metals during the CO₂ reduction, persisting “+1”? Are there any other roles of transition metals in CO₂ reduction?

RESPONSE: Thank the reviewer very much for raising the questions. From the current results, it is obvious that the most important role of the transition metals in CO₂ reduction is to provide active sites for the coordination of H₂O (**Figure 5a**). As reported in many previous reports [*J. Catal.* **1979**, *56*(3), 407-429; *Appl. Catal.* **1986**, *25*(1-2), 101-107; *Catal. Lett.* **1994**, *30*, 99-111; *Appl. Catal. A* **2001**, *218*(1-2), 235-240; *Surf. Sci.* **2007**, *601*(14), 3125-3132; *J. Electrochem. Soc.* **2011**, *158*(2), E45-E49; *Nat. Commun.* **2016**, *7*(1), 12123; *Nat. Catal.* **2018**, *1*(2), 103-110; *Nat. Commun.* **2018**, *9*(1), 3828], the monovalence state of transition metals such as Cu(I) species played a critical role in CO₂RR, and the stability of Cu(I) species would favor the related hydrogenation or reduction reactions [*J. Electrochem. Soc.* **2011**, *158*(2), E45-E49]. Moreover, direct evidence that Cu(I) species were stable in CO₂RR has been proved by Mistry *et al.* [*Nat. Commun.* **2016**, *7*(1), 12123]. Based on the current evidences, the oxidation state of transition metals (Cu, Ag, and Pd) in the reduction of CO₂ to CO mainly persisted +1, in good agreement with the above report. By the way, we have to admit that other oxidation states of transition metals such as Cu(0), Cu(II) are possibly involved in the reaction process (**Fig. S27c,d**), but no direct evidence could prove this point in the current condition. Based on this, the related description has been made as “...Despite this, the oxidation state of copper persisted +1, in good agreement with the previous report⁷⁷...” in the 3rd paragraph of Page 9.

For the roles of transition metals such as Cu(I) species, they not only provided active sites for coordination with H₂O, but also offered opportunity to charge neutral species such as CO₂ and CO for favorable mass spectrometric analysis by the forms of [Cu(CO₂)]⁺ and [Cu(CO)]⁺. The corresponding description has been made up as “...The above results also indicated that the coordinated H₂O on the Cu⁺, rather than free H₂O, played crucial roles for the reduction of CO₂ to CO. The transition metals such as Cu⁺ not only provided efficient active sites for the generation of [Cu(H₂O)]⁺ or other H₂O-based metal complex ions, but also offered opportunity to charge neutral species such as CO₂ and CO for favorable mass spectrometric analysis by the forms of [Cu(CO₂)]⁺ and [Cu(CO)]⁺ (Figs. 1 and 3)...” in the

2nd paragraph of Page 8.

7. In Figure 4, the number of coordinated H₂O has less effect on CO generation. If the [⁶³Cu(H₂O)]⁺ was the active site for CO₂ reduction to CO, two H in coordinated H₂O were needed and the O in coordinated H₂O was in CO. Then the pathways of O in CO₂ was not documented.

RESPONSE: Thank the reviewer so much for the comment. The reaction pathway between [⁶³Cu(H₂O)]⁺ and CO₂ by formation of CO was demonstrated in **Figure 5a**, namely “...*This process started with the formation of [Cu(H₂O)]⁺, which interacted with CO₂ to generate [Cu(H₂O)(CO₂)]⁺. As aforementioned, H₂CO₃ formed after the reaction of CO₂ and H₂O, thereby the occurrence of transition from [Cu(H₂O)(CO₂)]⁺ to [Cu(H₂CO₃)]⁺, which was similar to the direct CO₂ capture and conversion to fuels over magnesium nanoparticles.⁵⁴ Subsequent H⁺/e⁻ transfer reactions^{46,48,70} led to the generation of [Cu(CO)]⁺, along with the elimination of two H₂O molecules. In the CO₂RR, H⁺ ions originated from the dissociation of H₂O/H₂CO₃ or added acid, and the necessary electrons involved in the reduction of CO₂ to CO were likely from the generated H₂CO₃ (Fig. S27 and related discussion). Upon the CO₂ reduction, the O atom in [Cu(H₂O)]⁺ combined with the C atom in CO₂ by formation of CO, whereas the two O atoms in CO₂ were eliminated by loss of two H₂O molecules...*” as described in the 3rd paragraph of Page 9. From the above, the O atoms in CO₂ was eliminated by loss of two H₂O molecules through H⁺/e⁻ transfer reactions. Further detailed DFT microscopic reaction pathway is shown in **Figure 5b**, and the corresponding discussion is given in Pages 9-10. The content could be useful to paint the pathways of O in CO₂ reduction to CO.

Fig. 5 | The mechanism of CO₂ reduction to CO in the presence a Cu(I) catalyst and water. a, Schematic representation of the reduction of CO₂ to CO according to experimental observations. **b,** The DFT microscopic reaction pathway [the singlet reactants of ¹Cu⁺ + ¹H₂O + ¹CO₂ (¹ISO, 0.00) was taken to be zero as a reference], demonstrating the thermodynamic and kinetic feasibility of the suggested pathway.

Reviewer #3 (Remarks to the Author)

Comments:

Zheng *et al.* have considered the comments by this reviewer and implemented the necessary changes. One comment to address is that, while the thermal heterogeneous catalysis test was done using various water partial pressure, the comparison between the total conversion with and without H₂O dosage is missing. A graph on CO₂ conversion under both conditions (with and without steady state dosage of H₂O) should be added to the manuscript to clearly show the effect of water addition on the overall performance of the catalyst.

RESPONSE: We appreciate the time and effort that you dedicated to providing insightful comments and valuable improvements to our work. According to the suggestions, we have added a graph comparison on CO₂ conversion with and without steady state dosage of H₂O. The related result is shown in **Fig. S6c**, which may be clear to demonstrate the effect of water addition on the overall performance of the employed catalyst. The corresponding description has also been added in the discussion of **Fig.S6c** in the Supporting Information. Once again, we thank the reviewer for the valuable comments.

Fig. S6 | Effect of H₂O pressure on the reduction of CO₂ to CO using a water-gas shift reaction (WGSR) apparatus equipped with gas chromatography: (a) WGSR apparatus. (b) Effect of H₂O pressure on CO generation (reaction temperature: 230 °C; $p(\text{CO}_2) = 0.3$ MPa, $p(\text{H}_2) = 2.7$ MPa, and $p(\text{H}_2\text{O})$ was controlled by the flow rate of syringe pump; $n = 3$). (c) Comparison of CO₂ conversion efficiency in the absence and presence of optimized H₂O content (50 kPa).

REVIEWER COMMENTS 
Reviewer #2 (Remarks to the Author):

Please refer to the review attachment.

For Reviewer #2 (Remarks to the Author)

Comments:

In the response letter, the authors provided some additional supporting data and their answers to support the conclusion that “the coordinated H₂O promoted CO₂ adsorption and subsequent reduction, and the O atom originated from the coordinated H₂O on catalysts, rather than CO₂ itself”. Unfortunately, my concerns were still not resolved, and to some extent, the authors avoided discussing certain pivotal questions. There existed overclaim in this manuscript. Given the uncertainty listed below, the reviewer would not recommend this article for publication in *Nature Communications*.

RESPONSE: The authors thank very much for the reviewer’s valuable comments and criticism for improving the strength of this study. According to the insightful suggestions, the replies to the raised concerns are presented below.

The specific comments are as follows:

1. The authors did not mediate my concern regarding “The gap between TSQ mass spectrometer and realistic thermal or electrochemical catalysis”. In the response letter, the authors just listed the reaction conditions of different systems. In reviewer’s opinion, the observed results of CO₂ reduction in this triple-stage quadrupole mass spectrometer cannot be simply extrapolated to realistic heterogeneous catalysis (electrochemical CO₂ reduction or reversed water-gas shift reaction) due to the significant differences in reaction microenvironment. The present work investigated a very specific system (ionized metal reacting with H₂O and CO₂), unless its extension to heterogeneous catalysis could be well justified (with a sound instruction that does not mislead the readers).

RESPONSE: Thank the reviewer very much for the comment. We agree with the reviewer’s opinion that there are gaps between TSQ mass spectrometer and realistic thermal or electrochemical catalysis due to significant differences in reaction microenvironment. However, the essence of CO₂RR is the reduction of CO₂ at the surface of catalysts. For the CO₂ reduction in our developed TSQ mass spectrometer, it was the reaction between [Cu(H₂O)]⁺ and CO₂ in the gas phase without the interference of other complex factors. For the realistic heterogeneous catalysis, including electrochemical CO₂ reduction or reversed water-gas shift reaction, the CO₂ reduction to CO was affected by many parameters (e.g., solvent, gas circuit system, and catalyst composition). However, the reaction essence was still the interactions of CO₂ and metal-based catalysts. To get convincing evidence, three series of experiments were carried out to prove that the results from the triple-stage quadrupole mass spectrometer could be extended to heterogeneous catalysis.

For the first experiment, *in situ* diffuse reflectance infrared Fourier transform spectroscopy (*in situ* DRIFTS) was used to confirm the role of H₂O in CO₂RR at the temperatures of 150°C and 250°C (Response to Comment 10 raised by Reviewer #3 in the first review of this work). The detailed experimental procedure has been given on Page 12. **Figs. S7a** and **S7b** show the experimental results using Cu/ γ -Al₂O₃ as catalyst under the reaction temperatures of 250°C and 350°C, respectively. For both reaction

temperatures, the *in situ* DRIFT spectra had an analogous pattern by adding H₂O into the reaction system. Namely, after introducing CO₂ into the system (without H₂O), the absorption peak intensity of bicarbonate on γ -Al₂O₃ (1230 cm⁻¹, 1439 cm⁻¹, and 1670 cm⁻¹) demonstrated a gradually increasing trend and reached a plateau. Simultaneously, the absorption peak of bidentate formate (1602 cm⁻¹) on γ -Al₂O₃ steadily emerged, but this type of absorption peak on Cu (1649 cm⁻¹) was not obvious. However, after H₂O was introduced to the reaction system, on the one hand, the absorption peak of bidentate formate on γ -Al₂O₃ (1379 cm⁻¹, 1398 cm⁻¹, and 1602 cm⁻¹) gradually increased, along with the gradual decay till disappearance of the absorption peak of bicarbonate occurred at 1230 cm⁻¹ and 1439 cm⁻¹. More importantly, the absorption peak of bidentate formate on Cu (1649 cm⁻¹) exhibited a gradually increasing trend. These results indicated that H₂O played a crucial role in the generation of formate on both Cu and γ -Al₂O₃. According to our

Fig. S7 | *In situ* DRIFT spectra of the resulting products on Cu/ γ -Al₂O₃ with addition of H₂O into the reaction system of CO₂ and H₂ at different reaction temperatures: (a) 150 °C @Cu/ γ -Al₂O₃, (b) 250 °C @Cu/ γ -Al₂O₃, and (c) 50 °C @Pt/ γ -Al₂O₃ (flow rate of 5% CO₂/95% Ar: 4 mL min⁻¹; flow rate of H₂: 3 mL min⁻¹).

discussion on **Fig. 5a**, formate was a critical intermediate in the reduction of CO₂ to CO, which indirectly confirmed the critical role of H₂O in CO₂RR. Despite this, no obvious absorption peaks of CO adsorbed on either γ -Al₂O₃ or Cu were observed, due to its fast desorption rate or weak adsorption on both supports under the current conditions (*Catal. Sci. Technol.* **2013**, 3, 767-778).

To get more compelling results on the effect of H₂O in CO₂RR, Pt/ γ -Al₂O₃ was employed as a catalyst to perform the experiments. As shown in **Fig. S7c**, after introducing H₂O into the reaction system, the absorption peak of CO adsorbed on Pt (2000 cm⁻¹ and 2061 cm⁻¹) demonstrated a gradual increasing pattern with extending the reaction period. Such a result indicated that the introduced H₂O was indeed favorable to the generation of CO in CO₂RR.

For the second experiment, we employed an *online* differential electrochemical mass spectrometer (**Fig. S18**) and isotope-labeling H₂¹⁸O to monitor the CO₂RR using Au, Ag, and Pd as working electrodes at room temperature. The detailed experimental procedure has been given on Pages 12-13 of the main text. The related results are shown in **Figs. S19-S21**. In contrast to the system with H₂¹⁶O (**Figs. S19c, S20c, and S21c**), a considerable amount of C¹⁸O (m/z 30) was generated in the presence of H₂¹⁸O (**Figs. S19f, S20f, and S21f**), indicating that H₂O was the oxygen source of resulting CO from CO₂ reduction, in good agreement with our mass spectrometric results. The corresponding description has been made as “...The reaction products were monitored using an online differential electrochemical mass spectrometer (Fig. S18). In contrast to the system of H₂¹⁶O, a considerable amount of C¹⁸O (m/z 30) was generated in the presence of H₂¹⁸O (Figs. S19-S21). This further indicated that H₂O was the oxygen source of resulting CO from CO₂ reduction and provided solid evidence on the generalization of the current finding to related reactions in the condensed phase...” in the 2nd paragraph of Page 7.

Fig. S18 | Differential electrochemical mass spectrometer for the electrochemical reduction of CO₂ to CO and *in situ* mass spectrometer monitoring of resulting reaction products.

For the third experiment, we employed a water-gas shift reaction (WGS) apparatus (**Fig. S6a**) to investigate the effect of H₂¹⁶O and H₂¹⁸O on the reduction of CO₂ to CO, and commercial available Cu/ZnO/Al₂O₃ particles were used as catalysts. First, we studied the effect of H₂O on the reaction efficiency of CO₂ reduction to CO. It was found that with the increase of H₂O partial pressure, the regeneration amount of CO demonstrated a first increasing trend followed by a declining one, in which it gave the optimal performance

when the H₂O partial pressure in the reaction system was 50 kPa (Figs. S6b and S6c). Second, the reaction products were also collected and measured *off-line* with an Agilent Technologies 7890B gas chromatograph (GC) equipped with a GS-CarbonPLOT capillary column and an Agilent Technologies 5977A mass spectrometer (MS). The detailed procedures have been given in the 3rd paragraph of Page 12. Figs. S6d and S6e show the corresponding chromatograms after GC separation. Due to the limited separation capacity of GS-CarbonPLOT capillary column in a manual injection mode, the peaks of air (N₂ and O₂) and CO could not be resolved well, but both could be observed clearly. Subsequently, we identified the components of CO from the reaction systems of H₂¹⁶O and H₂¹⁸O using mass spectrometry. As demonstrated in Figs. S6f and S6g, besides other peaks resulting from N₂/CO (m/z 28) and O₂ (m/z 32) owing to the limited separation between air and CO, a more abundant peak of m/z 30 appeared for the product in the presence of H₂¹⁸O than that in the presence of H₂¹⁶O. According to the current reaction system, the peak of m/z 30 could be assigned to C¹⁸O from the interaction of CO₂ and H₂¹⁸O. More importantly, after comparing the peak intensity, it is apparent that an improvement of 46.9-fold in peak

Fig. S6 | Effect of H₂O pressure on the reduction of CO₂ to CO using a water-gas shift reaction (WGS) apparatus equipped with gas chromatography: (a) WGS apparatus. (b) Effect of H₂O pressure on CO generation (reaction temperature: 230 °C; p(CO₂) = 0.3 MPa, p(H₂) = 2.7 MPa, and p(H₂O) was controlled by the flow rate of syringe pump; n = 3). (c) Comparison of CO₂ conversion efficiency in the absence and presence of optimized H₂O content (50 kPa). Chromatograms of the collected gas samples prepared in the presence of 50 kPa of (d) H₂¹⁶O and (e) H₂¹⁸O. Mass spectra of CO products prepared in the presence of 50 kPa of (f) H₂¹⁶O and (g) H₂¹⁸O, corresponding to CO peaks in (d) and (e), respectively. (h) Comparison of the peak intensity of m/z 30 observed in (f) and (g).

intensity of m/z 30 was observed for the system of $H_2^{18}O$ than that in the presence of $H_2^{16}O$ (**Fig. S6h**). These results not only suggest that the peak m/z 30 was not from the background, but also brought further evidence that the O atom in resulting CO originated from the involved H_2O .

From the above, it is apparent that although much gap exists between TSQ mass spectrometer and *online* differential electrochemical mass spectrometer/*in situ* diffuse reflectance infrared Fourier transform spectroscopy/reversed WGSR, the conclusions from them are the same: (i) H_2O plays an essential role in CO_2RR ; (ii) The O atom in resulting CO from CO_2 reduction originates from H_2O . This case persuades us to trust the validity of the mass spectrometric results and its probable extension to realistic heterogeneous catalysis. Based on the above results, the reaction mechanism of the CO_2 reduction to CO was investigated in detail, along with the computational calculations with DFT. The data presented above allowed us to conclude that the O atom in resulting CO from the reduction of CO_2 originated from the involved H_2O , rather than CO_2 itself. Hope that the above fact can resolve the raised concern.

2. The authors carried out the realistic thermal catalysis experiment using water-gas shift reaction (WGSR) apparatus and try to support their conclusions. Unfortunately, the authors also acknowledge that the results just revealed the effect of H_2O content on the reduction of CO_2 to CO but did not bring solid evidence for the conclusion “O atom originated from the coordinated H_2O on catalysts, rather than CO_2 itself.” Regarding the H_2O effect on CO_2 hydrogenation, many experimental and theoretical studies have been investigated (*J. Catal.* 2011, 281, 199; *J. Catal.* 2020, 383, 283; *Chem* 2019, 6, 419; *J. Catal.* 2018, 367, 194).

RESPONSE: The authors thank very much to the reviewer’s valuable comments for improving the strength of the study. According to the insightful suggestions, it is indeed that in the previous versions of this work, we just provided the H_2O effect on CO_2 hydrogenation, but did not bring solid evidence to prove that the source of O atom in resulting CO was from involved H_2O using water-gas shift reaction (WGSR) apparatus. To make up such an experiment, we performed the corresponding $H_2^{18}O$ -labeling experiments in tandem with *off-line* GC-MS analysis as the above **Response to Comment 1**. As shown in **Fig. S6f-h**, in comparison with the system of $H_2^{16}O$, an abundant peak of m/z 30 ($C^{18}O$) emerged in the mass spectrum as $H_2^{18}O$ was introduced into the reaction of CO_2 to CO, indicating that the involved H_2O was indeed the oxygen source of resulting CO from CO_2 reduction. These results brought evidence for the conclusion of “O atom originated from the coordinated H_2O on catalysts, rather than CO_2 itself” from our mass spectrometric observation. The related discussion has been made as “Moreover, $H_2^{18}O$ -labeling experiments in tandem with off-line GC-MS analysis were carried out to confirm the source of O atom in resulting CO from the reversed WGSR. As shown in **Fig. S6f-h**, in comparison with the system of introducing $H_2^{16}O$, an abundant peak of m/z 30 ($C^{18}O$) emerged in the mass spectrum for the system of $H_2^{18}O$. This fact indicated that the involved H_2O was indeed the oxygen source of the resulting CO. More importantly, the observed results of CO_2 reduction in the TSQ mass spectrometer could be extrapolated to realistic heterogeneous catalysis” in the 2nd paragraph of Page 7.

3. The reviewer strongly doubts that the O exchange from the uncatalyzed CO₂ hydration resulted in the observed results “O atom originated from the coordinated H₂O on catalysts, rather than CO₂ itself”, because the kinetic of this reaction is fast. The detailed experiments and interpretation could be found in the **Figure C1**, coming from the paper

a)

b)

Figure C1. (a) Sketch of the possible oxygen-exchanging reactions between CO₂ and H₂O. Black is carbon, white is hydrogen, red is ¹⁶O and green is ¹⁸O. (b) The measured isotopic distribution of CO stripping experiment t (solid lines).

Reproduced with permission. Copyright 2021, Elsevier

“*Electrochimica Acta* 2021, 374, 137842”.

RESPONSE: We thank the reviewer for providing such an essential reference to unveil the origin of the O atom in the resulting CO, but we respectfully disagree with the concept that the O exchange from the uncatalyzed CO₂ hydration resulted in the observed results “O atom originated from the coordinated H₂O on catalysts, rather than CO₂ itself”.

First, we admitted the occurrence of the O exchange from the uncatalyzed CO₂ hydration in the provided reference (*Electrochimica Acta* 2021, 374, 137842). In the current reaction using WGS apparatus, we also observed the O exchange in the interaction of CO₂ and H₂¹⁸O. **Figs. R1a** and **R1b** show the gas chromatograms of the collected gas samples in the presence of 50 kPa of H₂¹⁶O and H₂¹⁸O, respectively, and the corresponding mass spectra of CO₂ were given in **Figs. R1c** and **R1d**. In contrast to the system with H₂¹⁶O (**Fig.**

R1c), abundant peaks of m/z 44 and 46 appeared in the mass spectrum as H_2^{18}O was introduced into the reaction system (**Fig. R1d**). They could be assigned to C^{16}O_2 and $\text{C}^{16}\text{O}^{18}\text{O}$, respectively, indicating that an exchange of O atom occurred when H_2^{18}O interacted with the reactant C^{16}O_2 . It could also be found that the peak intensity of $\text{C}^{16}\text{O}^{18}\text{O}$ was around 11.5% of that from C^{16}O_2 . From the mass spectra, no obvious reaction product of C^{18}O_2 (m/z 48) was observed, which revealed that its generation was less favorable than $\text{C}^{16}\text{O}^{18}\text{O}$ (m/z 46). From the above, we could conclude that although the O exchange from the uncatalyzed CO_2 hydration was favorable, only partial C^{16}O_2 could turn into $\text{C}^{16}\text{O}^{18}\text{O}$. In other words, in the reduction of CO_2 , both products of C^{16}O and C^{18}O could be generated if the source of O atom in resulting CO was from CO_2 itself. More importantly, the O atom in generated CO was mainly from the un-exchanged C^{16}O_2 due to its much more than $\text{C}^{16}\text{O}^{18}\text{O}$. Therefore, the peak intensity of m/z 28 (C^{16}O) should be more intensive than that of m/z 30 (C^{18}O) if the O atom in resulting CO was indeed from CO_2 itself. Due to the limited resolving ability of GC-MS for the peaks of air and CO (**Figs. R1a** and **R1b**) and unavoidable involvement of H_2^{16}O into the reaction system of WGSR apparatus (e.g., CO_2 purity, gas circuit system, and others), no direct evidence could confirm the above assumption after analyzing the products collected from the WGSR apparatus using GC-MS, even though H_2^{18}O was employed.

Fig. R1 | Effect of H_2O pressure on the reduction of CO_2 to CO using a water-gas shift reaction (WGSR) apparatus: GC chromatograms of the collected gas samples prepared in the presence of 50 kPa of (a) H_2^{16}O and (b) H_2^{18}O . Mass spectra of CO_2 products prepared in the presence of 50 kPa of (c) H_2^{16}O and (d) H_2^{18}O , corresponding to CO_2 peaks in (a) and (b), respectively [The peak of m/z 44 in (c) and (d) could be assigned as CO_2].

To address the above issues, the isotope-labeling experiments of C^{16}O_2 and C^{18}O_2 were performed in our developed TSQ mass spectrometer, in which C^{16}O_2 and C^{18}O_2 were, respectively, employed as reactants to react with $[\text{Cu}(\text{H}_2\text{O})]$. **Fig. 3d** shows the reaction between $[\text{Cu}(\text{H}_2^{16}\text{O})]$ and C^{18}O_2 . If a quick exchange between H_2^{16}O and C^{18}O_2 occurred, $\text{C}^{16}\text{O}_2/\text{C}^{16}\text{O}^{18}\text{O}/\text{C}^{18}\text{O}_2$ -related products ($[\text{Cu}(\text{C}^{16}\text{O}_2)]^+$: m/z 107, $[\text{Cu}(\text{C}^{16}\text{O}^{18}\text{O})]^+$: m/z 109, $[\text{Cu}(\text{C}^{18}\text{O}_2)]^+$: m/z 111) and $\text{C}^{16}\text{O}/\text{C}^{18}\text{O}$ -related products ($[\text{Cu}(\text{C}^{16}\text{O})]^+$: m/z 91, $[\text{Cu}(\text{C}^{18}\text{O})]^+$: m/z 93) could be observed. However, only C^{18}O_2 (m/z 111) and C^{16}O -related product (m/z 91) emerged in the mass spectrum. The same case was observed for the reaction of $[\text{Cu}(\text{H}_2^{16}\text{O})]$, H_2^{16}O , and C^{18}O_2 (**Fig. 3e**). These results indicated that on the

one hand, a quick exchange between H_2^{16}O and C^{18}O_2 was less favorable in the TSQ mass spectrometer than that with WGS apparatus. On the other hand, no emergence of C^{18}O -related peak but only the appearance of C^{16}O -related peak in the mass spectra suggested that in the system of C^{18}O_2 , the ^{16}O atom in resulting C^{16}O was from the involved H_2^{16}O , rather than C^{18}O_2 itself. This was *“the O exchange from the uncatalyzed CO_2 hydration resulted in the observed results ‘O atom originated from the coordinated H_2O on catalysts, rather than CO_2 itself”*. Such a fact revealed that our statement of *“O atom originated from the coordinated H_2O on catalysts, rather than CO_2 itself”* was reasonable.

Fig. 3 | Isotope-labeling MS measurement results under different systems. a, $[\text{Cu}(\text{H}_2^{16}\text{O})]^+$ and C^{16}O_2 . b, $[\text{Cu}(\text{H}_2^{16}\text{O})]^+$, H_2^{16}O , and C^{16}O_2 . c, $[\text{Cu}(\text{H}_2^{16}\text{O})]^+$, H_2^{18}O , and C^{16}O_2 . d, $[\text{Cu}(\text{H}_2^{16}\text{O})]^+$ and C^{18}O_2 . e, $[\text{Cu}(\text{H}_2^{16}\text{O})]^+$, H_2^{16}O , and C^{18}O_2 ; f, $[\text{Cu}(\text{H}_2^{16}\text{O})]^+$, H_2^{18}O and C^{18}O_2 (gas circuit temperature: 280 °C; reaction pressure: 1.5 mTorr).

4. Based on the explanation of electron origin in this system, the reviewer thinks this reaction condition (280°C) is different from the realistic electrochemical CO_2 reduction (room temperature) and the extended electrochemical CO_2 reaction mechanism is unreasonable.

RESPONSE: The authors thank the reviewer for the valuable comments, but we respectfully disagree with the concept. Redox reactions are characterized by the actual or formal transfer of electrons between chemical species, and there is no exception for CO_2RR . It is indeed that the current reaction condition (280 °C) is different from the realistic electrochemical CO_2 reduction (room temperature), whereas the different reaction conditions could not exclude the involvement of electrons in the CO_2 reduction to CO using our mass spectrometer system. More importantly, the extended CO_2 reaction mechanism is based on the prior mass spectrometric results and the current mass spectrometric observation rather than the electrochemical CO_2RR .

It is well known that the Q2 region in the TSQ mass spectrometer (**Scheme 1**) is generally employed as a collision cell for the collision-induced dissociation (CID) of selected ions (https://en.wikipedia.org/wiki/Triple_quadrupole_mass_spectrometer). In the

current investigation, we employed the Q2 region as a unit for the molecule-ion reaction between Cu-based catalysts (e.g., $[\text{Cu}(\text{H}_2\text{O})]^+$) and CO_2 , and it involved the reactants of $[\text{Cu}(\text{H}_2\text{O})]^+$, Cu^+ , H_2O , and CO_2 in the Q2 region during the reaction/collision between $[\text{Cu}(\text{H}_2\text{O})]^+$ and CO_2 . Along with them, $[\text{Cu}(\text{H}_2\text{CO}_3)]^+$ or H_2CO_3 would be generated upon the collision between $[\text{Cu}(\text{H}_2\text{O})]^+/\text{H}_2\text{O}$ and CO_2 (**Fig. S24**). Subsequently, on the one hand, H_2CO_3 dissociates through Eqs. (1) and (2), and the resulting H^+ ions would supply the necessary protons in **Fig. 5a** of the main text. On the other hand, many prior studies [*Chem. Rev.* **2002**, 102, 231–282; *Phys. Scr.* **2007**, 76 C56-C62; *J. Chem. Phys.* **2002**, 116, 6560-6566; *Chem. Phys. Lett.* **1984**, 106, 544-549; *Planet. Space Sci.* **1981**, 29, 735-739] have indicated that electrons could be dissociated from anions [e.g., PtBr_6^{2-} , $\text{Pd}(\text{CN})_4^{2-}$, $\text{Ru}_3\text{Co}(\text{CO})_{13}^-$, CO_3^- , and CO_2^-] in the CID of mass spectrometry. Based on the above fact, it is speculated that the required electrons in this study could be generated following Eqs. (3)-(5) in the dissociation of CO_3^{2-} .

If the above assumption was correct, O_2 should be produced. After examining the mass spectrum from the reaction of $[\text{Cu}(\text{H}_2\text{O})]^+$ and CO_2 , $[\text{Cu}(\text{O}_2)]^+$ ions, namely m/z 95 for $^{63}\text{Cu}(\text{O}_2)^+$ and m/z 97 for $^{65}\text{Cu}(\text{O}_2)^+$, were indeed captured as shown in **Fig. S28a** and **b**.

Fig. S28 | Mass spectra of the products by interaction between $[\text{Cu}(\text{H}_2\text{O})]^+$ and CO_2 and variation in the peak intensity of different Cu-based species with increasing collision energy: Mass spectra of the products by interaction (a) between $^{63}\text{Cu}(\text{H}_2\text{O})^+$ and CO_2 and (b) between $^{65}\text{Cu}(\text{H}_2\text{O})^+$ and CO_2 ; Variation in the peak intensity of different (c) ^{63}Cu -based and (d) ^{65}Cu -based species with increasing collision energy (gas circuit temperature: 280 °C; reaction pressure: 1.5 mTorr).

This fact suggested that in CO₂RR, the dissociation of H₂CO₃ was likely to be one of the electron sources for CO₂RR.

Although it also supplies electrons in the process of Cu⁺ to Cu²⁺, no direct experimental evidences could confirm the generation of Cu(II)-based species. As a result, the most possible electron source for the reduction of CO₂ to CO was from the dissociation of H₂CO₃ in the CID of mass spectrometry.

5. The concerns in questions 5, 6, and 7 in previous comments still exist.

RESPONSE: The authors thank very much to the reviewer's valuable comments. Below are the replies to the concerns in questions 5, 6, and 7.

Q5. In Page 3, the revised version, the sentence "Cu(I) is one of the most active species in CO₂RR" still lacks scientific rigor. The metallic Cu was already observed under reaction condition during electrochemical CO₂ reduction, while the surface oxide species usually resulted in the different product distributions (Birdja Y. et al., *Nat. Energy* 2019, 4, 732; Nitopi S. et. al., *Chem. Rev.* 2019, 119, 7610).

RESPONSE: Thank the reviewer very much for the comment. The mentioned sentence has been revised as "...Many previous studies^{8,26,55-59} have demonstrated that of the different oxidation states, Cu(I)-related species are one of the most important catalysts in CO₂RR..." in the 4th paragraph of Page 3. The revision is based on the following reasons.

(1) After studying the suggested references (*Nat. Energy* 2019, 4, 732; *Chem. Rev.* 2019, 119, 7610), it could be concluded that Cu(I)-based species was significant in CO₂ reduction. For example, Birdja et al (*Nat. Energy* 2019, 4, 732-745) stated that "...It is often assumed that during CO₂RR, the Cu_xO electrocatalyst should be completely converted to metallic Cu in CO₂RR. However, a residual oxide layer was shown to remain present on the surface during CO₂RR and the surface oxide and OD metallic layer were proposed as key reaction sites for catalysis¹²⁵. The formation of C3-C4 was also reported, supposedly due to the synergistic effect between Cu₂O and C1 adsorption, which resulted in a higher population of Cu⁺ species..." in the left panel of Page 738. From the above discussion, it is obvious that although the metallic Cu was already observed under the reaction condition during electrochemical CO₂ reduction, the most important active species for CO₂RR were still the Cu(I)-based catalysts.

(2) Up to now, much effort has been focused on the subject of Cu-related materials as active species for CO₂RR [e.g., *J. Catal.* 1979, 56(3), 407-429; *Nat. Commun.* 2016, 7(1), 12123; *Nat. Catal.* 2018, 1(2), 103-110; *Nat. Commun.* 2018, 9(1), 3828]. Among diverse Cu-based species, Cu(I) species have been proved playing a crucial role in determining the CO₂RR efficiency.

Q6. In the "rebuttal letter", the authors replied that the most important role of the transition metals in CO₂ reduction is to provide active sites for the coordination of H₂O. CO₂ reduction reaction occurs on various metals even over the *p*-block elements. What is the oxidation state of transition metals during the CO₂ reduction, persisting "+1"? Are there any other roles of transition metals in CO₂ reduction?

RESPONSE: To our understanding, there are two concerns in this comment. The replies to them are presented below.

- (1) According to the current experimental results (**Figs. 1 and 3**), the oxidation state of transition metals (Cu, Ag, and Pd) in the reduction of CO₂ to CO persisted +1, in good agreement with the previous reports [*J. Catal.* **1979**, 56(3), 407-429; *Appl. Catal.* **1986**, 25(1-2), 101-107; *Catal. Lett.* **1994**, 30, 99-111; *Appl. Catal. A* **2001**, 218(1-2), 235-240; *Surf. Sci.* **2007**, 601(14), 3125-3132; *J. Electrochem. Soc.* **2011**, 158(2), E45-E49; *Nat. Commun.* **2016**, 7(1), 12123; *Nat. Catal.* **2018**, 1(2), 103-110; *Nat. Commun.* **2018**, 9(1), 3828]. Based on this fact, the related description has been made as “...Despite this, the oxidation state of copper persisted +1, in good agreement with the previous report”...” in the 3rd paragraph of Page 9.
- (2) The evidence from the current study (**Figs. 1 and 3**) indicated that transition metals such as Cu(I) species not only provided active sites for coordination with H₂O, but also offered an opportunity to charge neutral species such as CO₂ and CO for favorable mass spectrometric analysis by the forms of [Cu(CO₂)]⁺ and [Cu(CO)]⁺. Based on the above fact, the roles of transition metals in CO₂ reduction have been described as “...The transition metals such as Cu⁺ not only provided efficient active sites for the generation of [Cu(H₂O)]⁺ or other H₂O-based metal complex ions, but also offered opportunity to charge neutral species such as CO₂ and CO for favorable mass spectrometric analysis by the forms of [Cu(CO₂)]⁺ and [Cu(CO)]⁺ (Figs. 1 and 3)...” in the 2nd paragraph of Page 8. Although we tried to figure out more information from the current system, no other roles of transition metals were observed in the investigation.

Fig. 5 | The mechanism of CO₂ reduction to CO in the presence a Cu(I) catalyst and water. a, Schematic representation of the reduction of CO₂ to CO according to experimental observations. b, The DFT microscopic reaction pathway [the singlet reactants of ¹Cu⁺ + ¹H₂O + ¹CO₂ (¹ISO, 0.00) was taken to be zero as a reference], demonstrating the thermodynamic and kinetic feasibility of the suggested pathway.

Q7. In Figure 4, the number of coordinated H₂O has less effect on CO generation. If the [Cu(H₂O)]⁺ was the active site for CO₂ reduction to CO, two H in coordinated H₂O were needed and the O in coordinated H₂O was in CO. Then the pathways of O in

CO₂ was not documented.

RESPONSE: The pathway of O atoms in CO₂ has been documented as “...Upon the CO₂ reduction, the O atom in [Cu(H₂O)]⁺ combined with the C atom in CO₂ by the formation of CO, whereas the two O atoms in CO₂ were eliminated by loss of two H₂O molecules. The released H₂O molecules could combine with Cu⁺ to form [Cu(H₂O)]⁺ for the next cycle of CO₂ reduction...” as described in the 3rd paragraph of Page 9, and the pathway of O in CO₂ has also been depicted in **Fig. 5a**.